# Bivalent mRNA booster encoding virus-like particles elicits potent polyclass receptor-binding domain antibodies in pre-vaccinated mice

Chengcheng Fan[1†], Alexander A Cohen[1†], Kim-Marie A Dam[2†], Annie V Rorick[1‡], Ange-Célia I Priso Fils[1], Zhi Yang[1§], Priyanthi NP Gnanapragasam[1], Luisa N Segovia[1], Kathryn E Huey-Tubman[1], Woohyun J Moon[3], Paulo JC Lin[3], Pamela J Bjorkman[1*], Magnus AG Hoffmann[2,4*]

[1]Division of Biology and Biological Engineering, California Institute of Technology, Pasadena, United States; [2]Gladstone Institutes, San Francisco, United States; [3]Acuitas Therapeutics, Vancouver, Canada; [4]Department of Medicine, University of California, San Francisco, San Francisco, United States

*For correspondence:
bjorkman@caltech.edu (PJB);
magnus.hoffmann@gladstone.
ucsf.edu (MAGH)

†These authors contributed equally to this work

Present address: ‡Department of Biochemistry, University of Washington, Seattle, United States; §Department of Molecular and Cell Biology, University of California, Berkeley, Berkeley, United States

## eLife Assessment

This **useful** and interesting study provides evidence that EABR mRNA is at least as effective as standard S mRNA vaccines for SARS-CoV-2. The authors provide **convincing** justification for the conclusion that the inconsistent statistical significance for Omicron is likely due to immune imprinting or original antigenic sin. In this regard, the significance of the findings is stronger as it points to possible challenges for updated vaccine strategies in overcoming immune imprinting.

**Abstract** mRNA vaccines emerged as a leading vaccine technology during the COVID-19 pandemic. However, their sustained protective efficacies were limited by relatively short-lived antibody responses and the emergence of SARS-CoV-2 variants, necessitating frequent and variant-updated boosters. We recently developed the ESCRT- and ALIX-binding region (EABR) mRNA vaccine platform, which encodes engineered immunogens that induce budding of enveloped virus-like particles (eVLPs) from the plasma membrane, thereby resulting in presentation of immunogens on cell surfaces and eVLPs. Prior studies showed that spike (S)-EABR mRNA-LNP immunizations elicited enhanced neutralizing antibody responses against ancestral and variant SARS-CoV-2 compared with conventional S mRNA-LNP in naïve mice, but the effectiveness of S-EABR mRNA-LNP boosters in the context of pre-existing immunity has not been investigated. Here, we evaluated monovalent Wuhan-Hu-1 (Wu1) and bivalent (Wu1/BA.5) S-EABR mRNA-LNP boosters in mice pre-vaccinated with conventional Wu1 S mRNA-LNP. Compared to conventional S mRNA-LNP boosters, the EABR approach enhanced monovalent and bivalent mRNA-LNP booster-induced neutralizing responses against Omicron subvariants BA.1, BA.5, BQ.1.1, and XBB.1, with bivalent S-EABR mRNA-LNP consistently eliciting the highest titers. Epitope mapping of polyclonal antisera by deep mutational scanning revealed that bivalent S-EABR mRNA-LNP boosted diverse 'polyclass' anti-receptor-binding domain (RBD) responses, suggesting balanced targeting of multiple RBD epitope classes. In contrast, monovalent S, bivalent S, and monovalent S-EABR mRNA-LNP boosters elicited less diverse polyclonal serum responses primarily targeting immunodominant RBD epitopes. Cryo-electron microscopy (cryo-EM) structures demonstrated that bivalent mRNA immunizations promote S heterotrimer formation, potentially enhancing bivalent S-EABR mRNA-LNP booster-induced

antibody breadth and polyclass epitope targeting by activating cross-reactive B cells through intra-S crosslinking. These findings support the future design of bivalent or multivalent S-EABR mRNA-LNP boosters as a promising strategy to confer broader, and therefore potentially more durable, protection against emerging SARS-CoV-2 variants and other rapidly evolving viruses.

## Introduction

mRNA vaccines emerged as a leading platform for the rapid and scalable development of effective vaccines during the COVID-19 pandemic (*Corbett et al., 2020*). Currently approved mRNA vaccines encode the SARS-CoV-2 spike (S) trimer (*Zheng et al., 2022*), the main target for neutralizing antibodies (Abs) produced during natural infection (*Chen et al., 2023*). S-encoding mRNAs formulated in lipid nanoparticles (LNPs) are taken up by cells, which promotes translation and cell surface expression of S trimers to activate B cells (*Hogan and Pardi, 2022*). Translation of mRNA-encoded S protein within host cells also provides viral peptides for presentation on MHC class I molecules, which are recognized by cytotoxic T cells (*Painter et al., 2021*; *Oberhardt et al., 2021*).

Initial two-shot COVID-19 mRNA vaccination series became available in December 2020 after clinical studies demonstrated protection from severe disease after SARS-CoV-2 infection (*Polack et al., 2020*; *Baden et al., 2021*). However, the durability of mRNA vaccine-induced immune protection was limited by two factors. First, serum Ab levels contracted over several months (*Zhang et al., 2022*). Second, SARS-CoV-2 variants acquired mutations in the S protein, which reduced their sensitivity to vaccine-induced neutralizing Abs (*Planas et al., 2021*). As a result, a third dose of the original mRNA vaccine encoding the ancestral Wuhan-Hu-1 (Wu1) S protein was offered starting in fall 2021 (*Andrews et al., 2022b*).

The emergence of the Omicron BA.1 variant in November 2021 (*WHO, 2021*) sparked concern because of its levels of substitution in S, especially in the receptor-binding domain (RBD), resulting in effective evasion of Ab-mediated immunity (*Liu et al., 2022*; *Cele et al., 2022*) and increased rates of symptomatic infections in previously vaccinated individuals (*Andrews et al., 2022a*; *Tseng et al., 2022*). Although undetectable after the initial mRNA vaccine series, a third dose of the original mRNA vaccine elicited neutralizing Abs against BA.1, but BA.1 neutralizing titers were lower compared to titers against the ancestral Wu1 variant (*Gruell et al., 2022*). The subsequent emergence of Omicron subvariants such as BA.5, BQ.1.1, and XBB.1, which exhibited progressively reduced sensitivities to vaccine-induced neutralizing Abs (*Wang et al., 2023c*), prompted the development of bivalent mRNA boosters, which included mRNAs encoding both the Wu1 and the BA.5 S proteins (*Scheaffer et al., 2023*). Preclinical and clinical studies found that bivalent boosters elicited modestly improved neutralizing Ab responses against Omicron subvariants compared to a fourth dose of the original monovalent vaccine (*Scheaffer et al., 2023*; *Khoury et al., 2023*; *Chalkias et al., 2022*; *Collier et al., 2023*; *Davis-Gardner et al., 2023*), although some studies found no differences between monovalent and bivalent boosters following initial vaccination with the original Wu1 S-encoding mRNA vaccine (*Wang et al., 2023a*; *Wang et al., 2023b*). Several studies (*Chalkias et al., 2022*; *Collier et al., 2023*; *Wang et al., 2023b*; *Alsoussi et al., 2023*; *Park et al., 2022*; *Carreño et al., 2023*) showed that the effectiveness of bivalent boosters was limited by immune imprinting, which occurs when prior exposure to an antigen shapes immunological responses to related antigens during subsequent exposures (*Cobey and Hensley, 2017*). Instead of generating de novo Omicron-specific immune responses, bivalent boosters primarily activated pre-existing cross-reactive memory B cells from previous vaccinations and/or infections (*Alsoussi et al., 2023*; *Park et al., 2022*; *Carreño et al., 2023*), leading to lower neutralizing activity against Omicron subvariants compared to the ancestral variant (*Chalkias et al., 2022*; *Collier et al., 2023*; *Wang et al., 2023b*).

Continued development of updated boosters in response to emerging variants was slowed by the need for additional clinical testing, regulatory approvals, and adjustments to the manufacturing process. As a result, bivalent boosters became available in fall 2022, almost 1 year after the emergence of Omicron (*Song et al., 2024*). These challenges highlight the urgent need for innovative vaccine strategies that achieve more lasting immune protection against rapidly evolving viruses such as SARS-CoV-2, thereby reducing the need for frequent and updated boosters.

We recently developed the ESCRT- and ALIX-binding region (EABR) mRNA vaccine platform that combines features of mRNA and protein nanoparticle (NP) vaccines through delivery of mRNA-encoded

self-assembling enveloped virus-like particles (eVLPs) (*Hoffmann et al., 2023*). eVLP assembly is achieved by fusing a short EABR sequence to the cytoplasmic tail of viral surface proteins, which recruits proteins from the <u>E</u>ndosomal <u>S</u>orting <u>C</u>omplex <u>R</u>equired for <u>T</u>ransport (ESCRT) pathway. Thus, EABR mRNA vaccines promote dual presentation of S trimers on cell surfaces and eVLPs that bud from the plasma membrane. Immunizations in mice showed that an EABR mRNA vaccine encoding a Wu1 SARS-CoV-2 S elicited superior neutralizing Ab potency and breadth against ancestral and Omicron SARS-CoV-2 variants compared to conventional mRNA and protein NP vaccines (*Hoffmann et al., 2023*). Similar to conventional mRNA vaccines, the EABR mRNA vaccine also induced potent T cell responses, consistent with the presence of S-specific cytotoxic T cells (*Hoffmann et al., 2023*). These findings suggest that the EABR mRNA vaccine approach has the potential to enhance monovalent or bivalent booster-induced Ab responses against Omicron subvariants in the context of pre-existing immunity induced by prior vaccination, infection, or both.

Here, we evaluated the ability of EABR mRNA boost immunogens to elicit potent and broad humoral immune responses against SARS-CoV-2 variants in pre-vaccinated mice. We demonstrate that the EABR mRNA vaccine approach improves monovalent and bivalent booster-mediated neutralizing Ab responses against Omicron subvariants compared to conventional mRNA boosters. Epitope mapping of polyclonal antisera further showed that a bivalent EABR mRNA booster elicited a more diversified anti-RBD IgG response than other tested boosting immunogens. These findings support the use of the EABR mRNA vaccine platform for the design of effective boosters that could confer more lasting protection against SARS-CoV-2 variants and potentially other rapidly evolving viruses. In addition, our work provides structural evidence that bivalent mRNA immunizations promote heterotrimer formation, which could enhance activation of cross-reactive B cells through intra-S crosslinking. These insights inform the future design of bivalent or multivalent COVID-19 boosters and highlight the need for further investigation into potentially beneficial effects of S heterotrimer formation on vaccine-induced immune responses against trimeric viral Ss.

## Results

To expand upon our previous characterizations of a monovalent SARS-CoV-2 S-EABR mRNA vaccine (*Hoffmann et al., 2023*), we evaluated the potential of the EABR mRNA vaccine approach for monovalent and bivalent SARS-CoV-2 immunizations in pre-vaccinated mice. Four cohorts of 10 BALB/c mice each initially received prime-boost intramuscular (IM) vaccinations on days 0 and 28 with 2 μg of a conventional mRNA vaccine encoding the ancestral S protein from the Wu1 variant, similar to the immunogens used in initially approved COVID-19 mRNA vaccines (Wu1 S mRNA-LNP) (*Figure 1A–C*). To simulate the timing of the third available COVID-19 mRNA vaccine dose in humans, booster immunizations were administered on day 230, nearly 7 months after the primary vaccination series, with each cohort receiving a different boost immunogen (*Figure 1A–C*). Consistent with the COVID-19 booster that became available in fall 2021 (*Andrews et al., 2022b*), cohort 1 received a third dose of 2 μg Wu1 S mRNA-LNP (*Figure 1B and C*). To evaluate whether formation of mRNA-encoded S-EABR eVLPs enhances booster immunogenicity in pre-vaccinated mice, cohort 2 received 2 μg Wu1 S-EABR mRNA-LNP (*Figure 1B and C*). Cohort 3 received a bivalent Wu1/BA.5 S mRNA-LNP booster (1 μg Wu1 S mRNA-LNP combined with 1 μg BA.5 S mRNA-LNP), which resembled the COVID-19 booster that became available in fall 2022 (*Song et al., 2024*; *Figure 1B and C*). Lastly, cohort 4 received a bivalent Wu1/BA.5 S-EABR mRNA-LNP booster (1 μg Wu1 S-EABR mRNA-LNP combined with 1 μg BA.5 S-EABR mRNA-LNP) to promote the formation of eVLPs that co-display Wu1 and BA.5 S trimers (*Figure 1B and C*). The Wu1 and BA.5 S-EABR constructs were designed as previously described by fusing an endocytosis-preventing motif (EPM), a short Gly-Ser linker, and an EABR sequence to the end of the cytoplasmic domain of Wu1 and BA.5 S trimers (*Hoffmann et al., 2023*; *Figure 1—figure supplement 1A*). The Wu1 and BA.5 S-EABR constructs induced similar levels of eVLP budding in transiently transfected HEK293T cells (*Figure 1—figure supplement 1B*).

Serum binding Ab responses were measured by enzyme-linked immunosorbent assay (ELISA) 4 and 18 weeks after the primary vaccination series (days 56 and 154) and 2 weeks after the booster immunizations (day 244). As previously observed, the initial Wu1 S mRNA-LNP vaccinations elicited high binding Ab titers against soluble Wu1 S-6P trimers (contain 6 proline substitutions to enhance expression and stability) (*Hsieh et al., 2020*), and as expected, these titers were similar for all cohorts on day 56 (*Figure 1D*). Robust binding Ab responses were also detected for all cohorts against Omicron

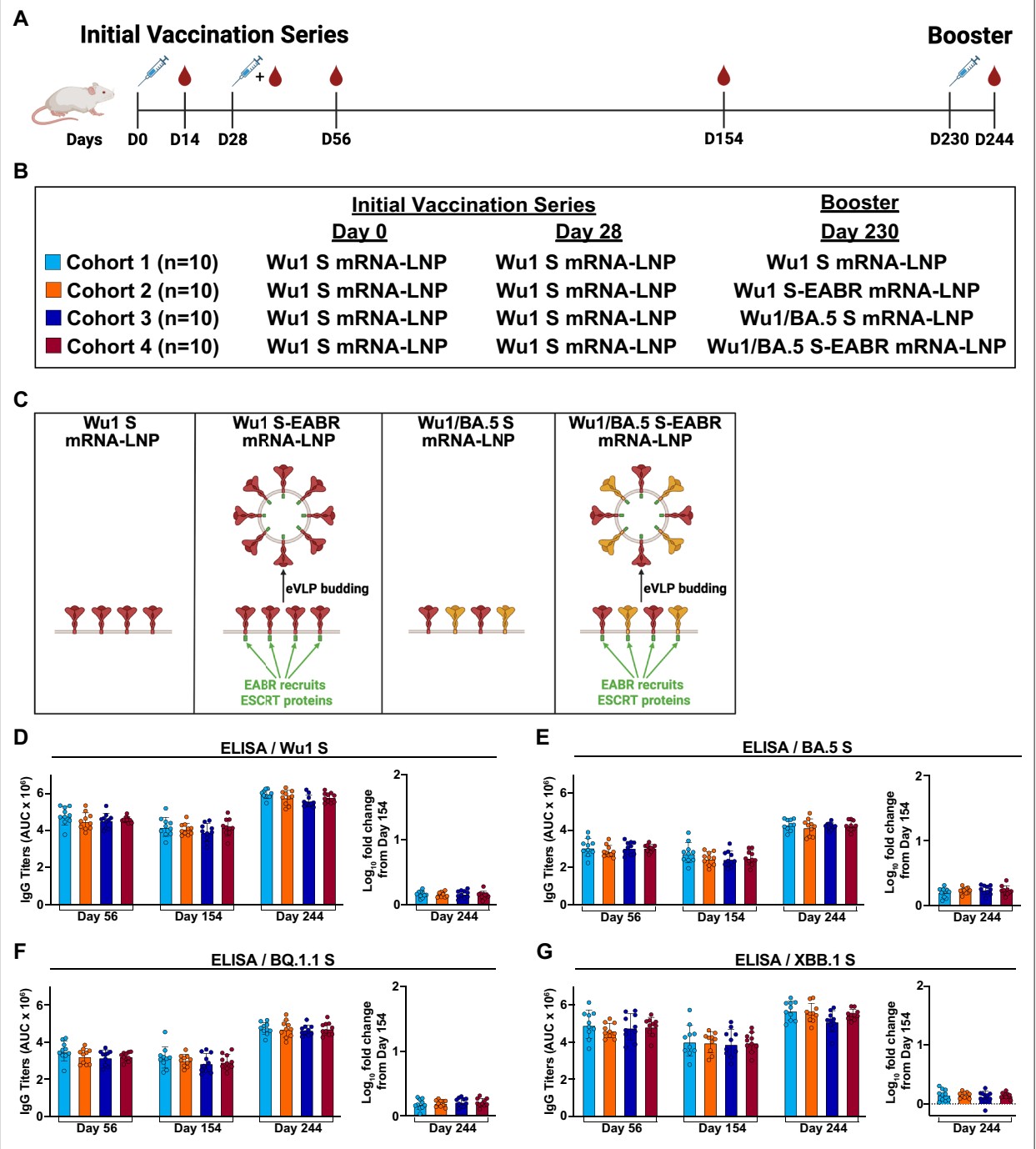

**Figure 1.** Monovalent and bivalent boosters elicit similar antibody (Ab) binding responses against variant S trimers in pre-vaccinated mice. (**A**) Immunization schedule in BALB/c mice. Blood samples were collected on days 14, 56, 154, and 244. This panel was created using BioRender.com. (**B**) Immunization regimens for each cohort. All cohorts were initially vaccinated with 2 µg of Wu1 S mRNA-LNP on days 0 and 28 by intramuscular (IM) injection. Pre-vaccinated mice received different boost immunogens on day 230 by IM injection. Cohorts 1–4 received 2 µg of monovalent Wu1 S mRNA-LNP (cyan), monovalent Wu1 S-EABR mRNA-LNP (orange), bivalent Wu1/BA.5 S mRNA-LNP (blue), or bivalent Wu1/BA.5 S-EABR mRNA-LNP (maroon), respectively. (**C**) Schematic showing the different boost immunogens that were administered on day 230. Wu1 S mRNA-LNP immunization results in cell surface presentation of membrane-anchored Wu1 S trimers. Wu1 S-EABR mRNA-LNP immunization results in dual presentation of Wu1 S trimers on cell surfaces and on enveloped virus-like particles (eVLPs) that are secreted after budding from the plasma membrane. eVLP formation is induced by EABR-mediated recruitment of host cell proteins from the ESCRT pathway. Bivalent Wu1/BA.5 S mRNA-LNP immunization results in cell surface presentation of membrane-anchored Wu1 and BA.5 S trimers. Wu1/BA.5 S-EABR mRNA-LNP immunization results in dual presentation of Wu1 and BA.5 S trimers on cell surfaces and eVLPs that bud from the plasma membrane. S trimers are shown as homotrimers for both monovalent and

*Figure 1 continued on next page*

*Figure 1 continued*

bivalent boost immunogens. This panel was created using BioRender.com. (**D–G**) Enzyme-linked immunosorbent assay (ELISA) data from indicated time points for antisera from individual mice (colored circles) presented as the geometric mean (bars) and standard deviation (horizontal lines) of area under the curve (AUC) values. ELISAs evaluated binding of purified Ss from the (**D**) Wu1, (**E**) BA.5, (**F**) BQ.1.1, and (**G**) XBB.1 variants. The $\log_{10}$-fold change in binding titers from day 154 to day 244 is also shown for each S (plotted as mean [bars] and standard deviation [horizontal lines]).

The online version of this article includes the following figure supplement(s) for figure 1:

**Figure supplement 1.** Wu1 and BA.5 S-EABR constructs induce efficient enveloped virus-like particle (eVLP) budding.

**Figure supplement 2.** Sarbecovirus receptor-binding domain (RBD) sequence conservation.

**Figure supplement 3.** Primary vaccinations and booster immunizations elicit potent anti-S2 binding responses.

BA.5, BQ.1.1, and XBB.1 S-6P trimers on day 56 (*Figure 1E–G*). On day 154, binding Ab levels against all variants dropped slightly, which was consistent between cohorts (*Figure 1D–G*). On day 244, all boost immunogens increased binding Ab titers against Wu1, BA.5, BQ.1.1, and XBB.1 S-6P trimers compared to titers measured on days 56 and 154, but no differences were observed between cohorts for all time points (*Figure 1D–G*).

## Bivalent S-EABR mRNA-LNP booster elicits potent Ab responses targeting Omicron RBDs

We next evaluated the impact of the different boost immunizations on Ab responses against the RBD, the primary target of neutralizing Abs (*Chen et al., 2023*). The initial Wu1 S mRNA-LNP prime-boost

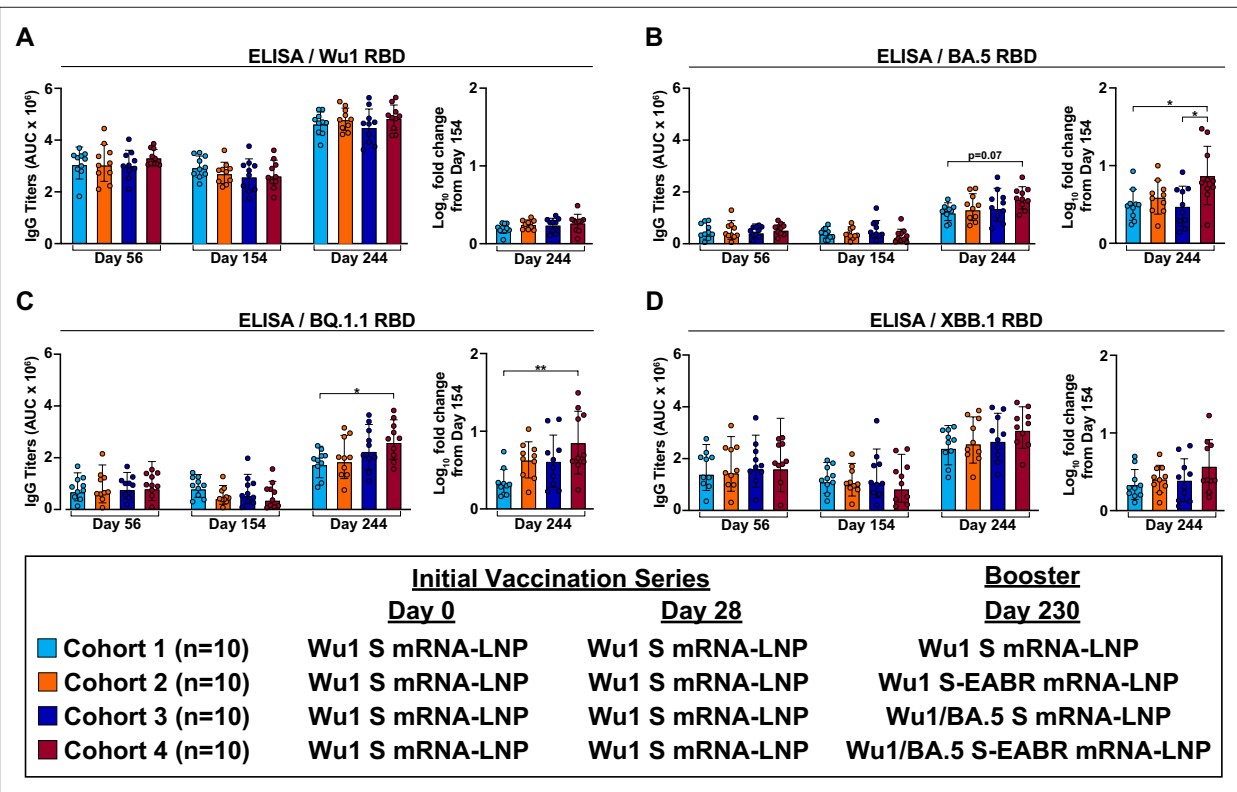

**Figure 2.** Bivalent S-EABR mRNA-LNP booster elicits potent antibody (Ab) responses targeting Omicron receptor-binding domains (RBDs). (**A–D**) RBD binding titers for antisera from pre-vaccinated mice that received booster immunizations with 2 µg of monovalent Wu1 S mRNA-LNP (cyan), 2 µg of monovalent Wu1 S-EABR mRNA-LNP (orange), 2 µg of bivalent Wu1/BA.5 S mRNA-LNP (blue), or 2 µg of bivalent Wu1/BA.5 S-EABR mRNA-LNP (maroon) (immunization schedule in *Figure 1A and B*). Enzyme-linked immunosorbent assay (ELISA) data are shown for indicated time points for antisera from individual mice (colored circles) presented as the geometric mean (bars) and standard deviation (horizontal lines) of area under the curve (AUC) values. ELISAs evaluated binding of RBDs from the (**A**) Wu1, (**B**) BA.5, (**C**) BQ.1.1, and (**D**) XBB.1 SARS-CoV-2 variants. The $\log_{10}$-fold change in binding titers from day 154 to day 244 is also shown for each RBD (plotted as mean [bars] and standard deviation [horizontal lines]). Significant differences between cohorts linked by horizontal lines are indicated by asterisks: $p<0.05$ = *, $p<0.01$ = **.

vaccinations elicited robust and similar Ab responses against the Wu1 RBD for all cohorts on day 56 (*Figure 2A*). The BA.5, BQ.1.1, and XBB.1 variants harbor multiple RBD substitutions, particularly within the class 1 and 2 epitopes at the tip of the RBD (*Barnes et al., 2020*; *Figure 1—figure supplement 2*), promoting effective escape from neutralizing Abs (*Wang et al., 2023c*). Compared to Wu1 RBD binding, Ab binding titers were weaker against the BA.5, BQ.1.1, and XBB.1 RBDs on day 56 (*Figure 2B–D*). The robust Ab binding against the BA.5, BQ.1.1, and XBB.1 S trimers (*Figure 1E–G*) in the absence of strong RBD responses was at least in part driven by potent responses against the S2 stem region of S (residues 686–1213) (*Figure 1—figure supplement 3*), which is relatively conserved in SARS-CoV-2 variants (*Li and Chang, 2023*; *Figure 1—figure supplement 2*). Similar to binding titers against S-6P trimers (*Figure 1D–G*), Wu1, BA.5, BQ.1.1, and XBB.1 RBD binding slightly dropped on day 154 without statistically significant differences between cohorts (*Figure 2A–D*). The booster immunizations increased Wu1 RBD binding on day 244 compared to day 56 and 154 levels, but no differences were detected between the four boosting immunogens (*Figure 2A*). In contrast, the bivalent Wu1/BA.5 S-EABR mRNA-LNP booster elicited the highest BA.5, BQ.1.1, and XBB.1 RBD responses on day 244 (*Figure 2B–D*), which were significantly higher than Wu1 S mRNA-LNP booster-induced responses against BQ.1.1 RBD (p=0.0499) (*Figure 2C*) and narrowly failed to reach statistical significance compared to Wu1 S mRNA-LNP booster-induced responses against BA.5 RBD (p=0.07) (*Figure 2B*). Compared to baseline titers measured prior to boost immunizations on day 154, the bivalent Wu1/BA.5 S-EABR mRNA-LNP immunization elicited significantly larger increases in BA.5 RBD binding than the monovalent Wu1 S mRNA-LNP (p=0.02) and the bivalent Wu1/BA.5 S mRNA-LNP boosters (p=0.01) (*Figure 2B*), as well as significantly larger increases in BQ.1.1 RBD binding than the Wu1 S mRNA-LNP booster (p=0.002) (*Figure 2C*). While overall titers were similar on day 244, the monovalent Wu1 S-EABR mRNA-LNP booster consistently induced larger increases in Ab binding to the BA.5, BQ.1.1, and XBB.1 RBDs than the monovalent Wu1 S mRNA-LNP booster (*Figure 2B–D*), although these differences failed to reach statistical significance. The monovalent Wu1 and bivalent Wu1/BA.5 S mRNA-LNP boosters elicited similar binding titers against the BA.5 and XBB.1 RBDs (*Figure 2B and D*), but the bivalent booster induced slightly higher responses and a larger increase compared to day 154 titers against BQ.1.1 RBD (not statistically significant) (*Figure 2C*). Overall, these results demonstrate that booster immunizations with the bivalent Wu1/BA.5 S-EABR mRNA-LNP elicited the highest Ab responses against Omicron RBDs in pre-vaccinated mice. Notably, post-boost Ab binding against Omicron RBDs remained lower than binding titers against Wu1 RBD in all cohorts, suggesting that booster responses were affected by immune imprinting induced by the primary vaccination series.

## Bivalent S-EABR mRNA-LNP booster elicits higher neutralizing responses against Omicron variants

The ability of boost immunogens to elicit neutralizing Ab responses in pre-vaccinated mice was evaluated by in vitro pseudovirus neutralization assays (*Crawford et al., 2020*; *Robbiani et al., 2020*). The primary vaccination series with Wu1 S mRNA-LNP elicited potent and consistent autologous geometric mean neutralizing titers as assessed by half-maximal serum inhibitory dilution ($ID_{50}$) values against Wu1 pseudovirus in all cohorts on day 56 (*Figure 3A*; *Supplementary file 1*). Neutralizing activity was >100-fold lower against the BA.5 variant and undetectable in most animals against the BQ.1.1 and XBB.1 variants on day 56 for all cohorts (*Figure 3B–D*; *Supplementary file 1*). On day 154, neutralizing titers against the Wu1 and BA.5 variants were consistent in all cohorts and slightly lower compared to day 56 titers (*Figure 3A and B*; *Supplementary file 1*). Neutralizing activity against the BQ.1.1 and XBB.1 variants was undetectable in all but one animal on day 154 (*Figure 3C and D*; *Supplementary file 1*). Booster immunizations on day 230 increased autologous neutralizing titers against Wu1 pseudovirus on day 244 compared to day 154 in all cohorts (*Figure 3A*; *Supplementary file 1*). Monovalent Wu1 S mRNA-LNP and S-EABR mRNA-LNP immunizations elicited the smallest and largest increases in Wu1 neutralization (4.2-fold versus 6.6-fold), respectively. All boost immunogens elicited increased BA.5 neutralizing activity on day 244 compared to day 56 and 154 titers (*Figure 3B*; *Supplementary file 1*). Boosting with monovalent Wu1 S-EABR mRNA-LNP induced 4.8-fold higher BA.5 neutralizing titers than a third Wu1 S mRNA-LNP immunization, but this difference did not reach statistical significance. Using the respective day 154 titers as baseline, the Wu1 S-EABR mRNA-LNP booster induced a 23.8-fold increase in BA.5 neutralization on day 244 compared to

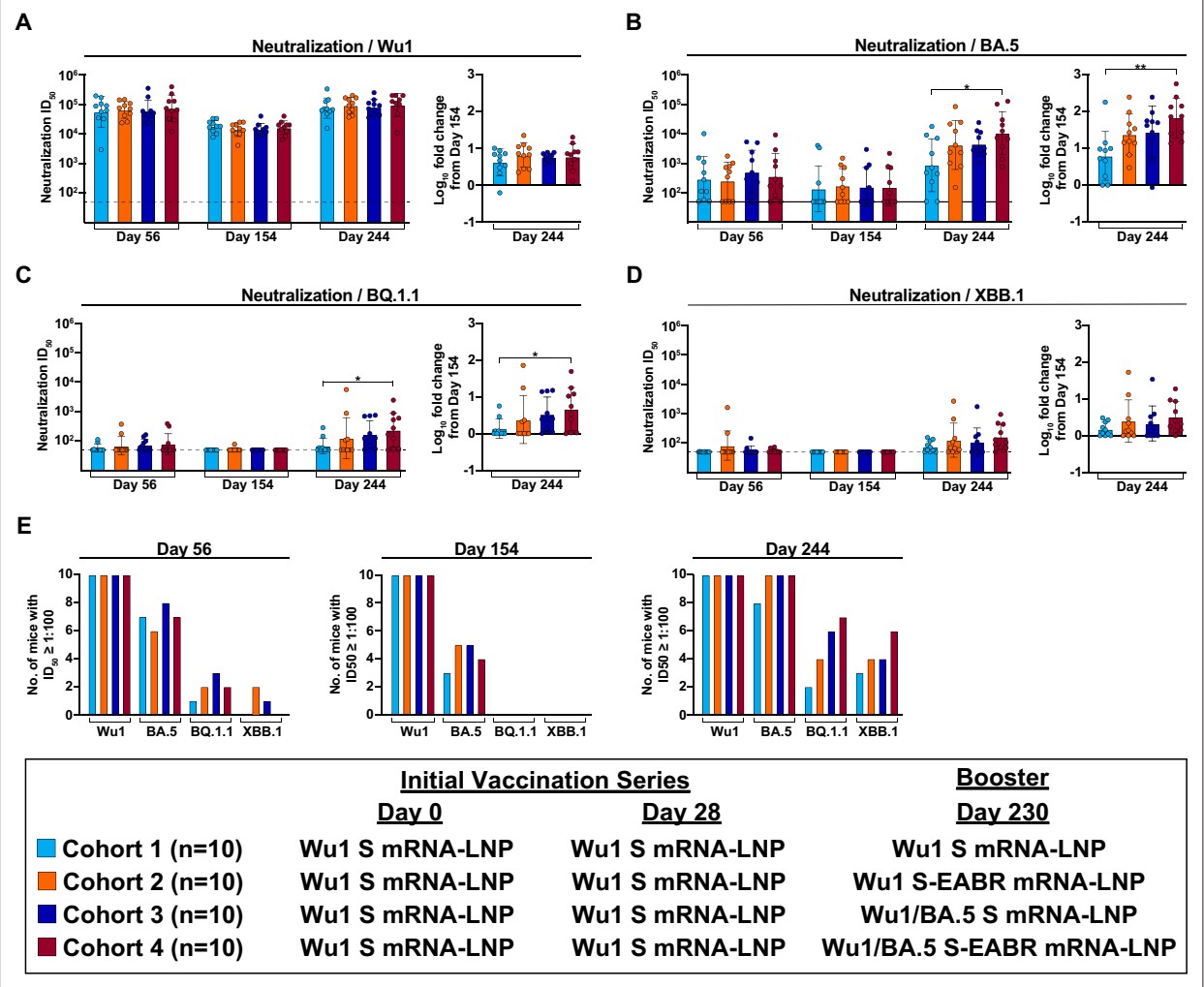

**Figure 3.** Bivalent Wu1/BA.5 S-EABR mRNA-LNP booster elicits higher neutralizing responses against Omicron subvariants. (A–D) Neutralization titers for antisera from pre-vaccinated mice that received booster immunizations with 2 µg of monovalent Wu1 S mRNA-LNP (cyan), 2 µg of monovalent Wu1 S-EABR mRNA-LNP (orange), 2 µg of bivalent Wu1/BA.5 S mRNA-LNP (blue), or 2 µg of bivalent Wu1/BA.5 S-EABR mRNA-LNP (maroon) (immunization schedule in *Figure 1A and B*). Neutralization data are shown for indicated time points for antisera from individual mice (colored circles) presented as the geometric mean (bars) and standard deviation (horizontal lines) of $ID_{50}$ values. Results are shown for (A) Wu1, (B) BA.5, (C) BQ.1.1, and (D) XBB.1 pseudoviruses. Dashed horizontal lines correspond to the background values representing the limit of detection for neutralization assays ($ID_{50}$=1:50). The $log_{10}$-fold change in neutralization titers from day 154 to day 244 is also shown for each pseudovirus (plotted as mean [bars] and standard deviation [horizontal lines]). Significant differences between cohorts linked by horizontal lines are indicated by asterisks: p<0.05 = *, p<0.01 = **. (E) Number of mice in each cohort that reached an $ID_{50}$ ≤ 1:100 against each variant on days 56 (left), 154 (middle), and 244 (right).

The online version of this article includes the following figure supplement(s) for figure 3:

**Figure supplement 1.** Monovalent and bivalent S-EABR mRNA-LNP boosters elicit higher increases in BA.5 neutralization compared to conventional S mRNA-LNP boosters.

**Figure supplement 2.** Monovalent Wu1 S-EABR mRNA-LNP booster elicits improved neutralizing responses against BA.1 variant.

**Figure supplement 3.** Depleting bivalent booster-induced antisera with Wu1 receptor-binding domain (RBD) reduces BA.5 neutralizing activity.

only a 6.3-fold increase detected after boosting with Wu1 S mRNA-LNP (not statistically significant) (*Figure 3B*; *Supplementary file 1*). Compared to day 56 baseline titers, the Wu1 S-EABR mRNA-LNP booster induced a significantly higher increase in BA.5 neutralization on day 244 than the conventional Wu1 S mRNA-LNP booster (16.2-fold versus 3.1-fold; p=0.048) (*Figure 3—figure supplement 1*; *Supplementary file 1*). The Wu1 S-EABR mRNA-LNP booster also elicited 3.4-fold higher neutralizing responses than the Wu1 S mRNA-LNP booster against the Omicron BA.1 variant on day 244 (p=0.03) (*Figure 3—figure supplement 2*). Based on day 154 titers, boosting with monovalent Wu1 S-EABR

mRNA-LNP induced significantly higher increases in BA.1 neutralization compared to monovalent Wu1 S mRNA-LNP (8.8-fold versus 2.5-fold; p=0.009) (*Figure 3—figure supplement 2*). Interestingly, the monovalent Wu1 S-EABR mRNA-LNP and the bivalent Wu1/BA.5 S mRNA-LNP boosters elicited similar BA.5 neutralizing responses on day 244 (*Figure 3B*; *Supplementary file 1*), suggesting that the EABR mRNA vaccine approach might reduce the need for updating boost immunogens to achieve robust Ab responses against emerging SARS-CoV-2 variants. The highest BA.5 neutralizing titers were elicited by the bivalent Wu1/BA.5 S-EABR mRNA-LNP booster, which were 12.3-fold higher than titers induced by the monovalent Wu1 S mRNA-LNP booster (p=0.02) and 2.4-fold higher than titers induced by the bivalent Wu1/BA.5 S mRNA-LNP booster (not statistically significant). Based on titers measured on day 154, the Wu1/BA.5 S-EABR mRNA-LNP booster induced a 67-fold increase in BA.5 neutralization, which was significantly higher than the increase elicited by boosting with monovalent Wu1 S mRNA-LNP (p=0.003).

While the BA.5 neutralizing activity induced by the monovalent Wu1 S and S-EABR mRNA-LNP boosters can likely be attributed to cross-neutralizing Abs that neutralize both the Wu1 and BA.5 variants, it is possible that the bivalent Wu1/BA.5 S and S-EABR mRNA-LNP boosters also elicited de novo strain-specific Abs that neutralize BA.5 but not Wu1. Day 244 serum samples from mice that received bivalent Wu1/BA.5 S or S-EABR mRNA-LNP boosters were depleted of Abs that bind to Wu1 RBD. Sera were passaged over a column presenting either a mock protein (HIV-1 gp120) as a control or the Wu1 RBD to remove cross-neutralizing anti-RBD Abs while retaining BA.5 RBD-specific Abs. Following depletions, residual neutralization activity was measured against the BA.5 variant. Interestingly, Wu1 RBD depletion resulted in 17- and 36-fold lower geometric mean neutralizing titers compared to mock depletion for the bivalent Wu1/BA.5 S and S-EABR mRNA-LNP boosters, respectively (*Figure 3—figure supplement 3*). In both cohorts, BA.5 neutralizing titers in all but two mice dropped below $ID_{50}$ values of 1:120 following Wu1 RBD depletion, consistent with the neutralizing activity in these samples being primarily driven by cross-neutralizing Abs targeting the RBD induced by the primary vaccination series. The potent residual neutralizing activity observed in two mice in each cohort after Wu1 RBD depletion implies either the presence of boost-induced de novo BA.5-specific RBD Abs or neutralizing non-RBD Abs that target another region of S, e.g., the N-terminal domain (*Wang et al., 2022*) or S2 (*Li and Chang, 2023*).

Booster immunizations had less pronounced effects on neutralization activity against the BQ.1.1 and XBB.1 variants. Against BQ.1.1 pseudovirus, the bivalent Wu1/BA.5 S-EABR mRNA-LNP booster elicited the most potent neutralizing activity on day 244, which was significantly higher than responses induced by a third immunization with the monovalent Wu1 S mRNA-LNP booster (p=0.0495) (*Figure 3C*; *Supplementary file 1*). The monovalent Wu1 S-EABR mRNA-LNP and bivalent Wu1/BA.5 S mRNA-LNP boosters elicited 1.8- and 2.5-fold higher BQ.1.1 titers than the monovalent Wu1 S mRNA-LNP booster, respectively, the latter of which narrowly failed to reach statistical significance (p=0.07). Based on day 154 BQ.1.1 neutralizing titers, the monovalent Wu1 S mRNA-LNP and bivalent Wu1/BA.5 S-EABR mRNA-LNP boosters induced the smallest and largest increases in neutralizing activity on day 244, respectively (1.4-fold versus 4.7-fold; p=0.0495). The monovalent Wu1 S-EABR mRNA-LNP and bivalent Wu1/BA.5 S mRNA-LNP boosters elicited 2.4- and 3.4-fold increases in neutralizing BQ.1.1 titers, respectively. Notably, day 244 neutralizing titers against BQ.1.1 were substantially lower than BA.5 neutralizing titers for all boost immunogens (*Figure 3B and C*; *Supplementary file 1*), consistent with previous reports (*Wang et al., 2023c*; *Miller et al., 2023*). Notably, Ab binding responses against the BA.5 and BQ.1.1 RBDs were similar (*Figure 2B and C*), implying that RBD binding responses are not always predictive of neutralizing activity against Omicron subvariants.

Neutralizing responses against the XBB.1 variant were slightly lower than titers against BQ.1.1 pseudovirus (*Figure 3D*; *Supplementary file 1*). Once again, the bivalent Wu1/BA.5 S-EABR mRNA-LNP booster elicited the highest XBB.1 neutralizing titers, which were 2.1-fold higher than titers induced by the monovalent Wu1 S mRNA-LNP booster (not statistically significant). Based on day 154 titers, the monovalent Wu1 and bivalent Wu1/BA.5 S-EABR mRNA-LNP boosters increased XBB.1 neutralization by 2.5- and 3.2-fold, respectively, while conventional monovalent and bivalent S mRNA-LNP boosters only induced 1.5- and 2.2-fold increases, respectively.

Notable differences were also observed in the number of mice in each cohort that reached robust neutralizing titers (defined as $ID_{50}$s exceeding 1:100) against Omicron subvariants following booster immunizations, titers that are likely important for the prevention of symptomatic infections (*Feng*

et al., 2021). The primary vaccination series with Wu1 S mRNA-LNP elicited robust neutralizing titers against the Wu1 variant in all mice on day 56 (*Figure 3E*). However, the initial Wu1 S mRNA-LNP vaccinations elicited robust titers in only 6–8, 1–3, and 0–2 of 10 mice against the BA.5, BQ.1.1, and XBB.1 variants on day 56, respectively. On day 154, $ID_{50}$s over 1:100 were measured for 3–5 mice in each cohort against BA.5, but for none of the animals against BQ.1.1 and XBB.1. While a third immunization with Wu1 S mRNA-LNP induced robust titers against BA.5 in only 8 of 10 mice on day 244, the monovalent Wu1 S-EABR mRNA-LNP booster induced robust titers against BA.5 in all 10 mice, including the 4 mice that did not have detectable BA.5 neutralizing activity on day 56 (*Figure 3E*). As expected, robust neutralizing titers against the BA.5 variant were also observed in all mice that received the bivalent Wu1/BA.5 S or S-EABR mRNA-LNP boosters. Against the BQ.1.1 and XBB.1 variants, a third immunization with Wu1 S mRNA-LNP induced robust neutralizing titers in only 2 and 3 mice, respectively (*Figure 3E*). In comparison, the bivalent Wu1/BA.5 S-EABR mRNA-LNP booster induced $ID_{50}$s over 1:100 in 7 and 6 mice against the BQ.1.1 and XBB.1 variants, respectively.

Overall, these results demonstrate that the EABR mRNA vaccine approach improves the neutralizing activity against Omicron subvariants induced by monovalent and bivalent boost immunogens in pre-vaccinated mice.

## Bivalent S-EABR mRNA-LNP booster induces polyclass Ab responses

Several distinct Ab-binding epitopes have been defined on RBDs, including the variable class 1 and 2 RBD epitopes and the more conserved class 3, 4, 1/4, and 5 RBD epitopes (*Barnes et al., 2020*; *Fan et al., 2022*; *Fan et al., 2025*; *Figure 1—figure supplement 2*). We determined which RBD epitopes were targeted by booster-induced Abs in pre-vaccinated mice using deep mutational scanning (DMS) of yeast display libraries derived from Wu1 (*Starr et al., 2020*) and XBB.1.5 RBDs (*Taylor and Starr, 2023*). In previous studies, we showed that equimolar mixtures of monoclonal Abs targeting distinct RBD epitopes produced less pronounced DMS escape peaks, which we defined as a polyclass Ab response (*Cohen et al., 2024*). This suggests that polyclonal serum that includes roughly equal distributions of Abs targeting multiple classes of RBD epitopes would exhibit relatively flat DMS profiles, while more defined DMS escape peaks would be observed for less diverse polyclonal serum responses containing Abs that primarily target particular RBD epitopes (*Cohen et al., 2024*). For this study, we used DMS escape fractions to define polyclass responses. Escape fractions for each RBD mutation range from 0 (no cells with this mutation are sorted into the serum Ab escape bin) to 1 (all cells with this mutation are sorted into the serum Ab escape bin) (*Greaney et al., 2021b*). Total escape peaks represent the sum of escape fractions for all mutations at a specific site (*Greaney et al., 2021b*). DMS profiles with total escape peaks below 0.5 at all sites were classified as polyclass responses. In contrast, DMS profiles with total escape peaks of 0.5–1, 1–2, or above 2 at one or more sites were classified as weak, moderate, or strong escape profiles, respectively, indicating weak, moderate, or strong targeting of particular RBD epitopes (*Supplementary file 2*). DMS was performed for 6 serum samples for each cohort, which were selected based on XBB.1 neutralization activity (1st, 2nd, 3rd, 5th, 6th, and 10th best neutralizer from each cohort). For the Wu1 RBD library, only 5 DMS profiles were obtained due to poor sequencing quality for one sample in each cohort.

Against the Wu1 RBD library, immunizations with the monovalent Wu1 S and S-EABR mRNA-LNP boosters or the bivalent Wu1/BA.5 S mRNA-LNP booster resulted in DMS profiles consistent with moderate targeting of the variable class 2 epitope primarily inducing escape at RBD residue 484 (*Figure 4A and B*). The bivalent Wu1/BA.5 S mRNA-LNP booster also elicited weak to moderate responses that mapped to the variable class 1 and class 3 RBD epitopes. In contrast, the bivalent Wu1/BA.5 S-EABR mRNA-LNP booster DMS profiles were consistent with polyclass Ab induction, suggesting a more diverse response targeting multiple RBD epitopes (*Figure 4A and B*). Notably, samples from 4 out of 5 mice in this cohort exhibited polyclass DMS profiles, whereas moderate to strong escape fractions for class 1, 2, or 3 epitopes were observed in 3–4 mice in each of the other cohorts (*Figure 4—figure supplement 1*; *Supplementary file 2*).

As expected, no class 1 and 2 DMS profiles were observed against the XBB.1.5 RBD library (*Taylor and Starr, 2023*) since these epitopes are highly substituted in Omicron subvariants (*Figure 1—figure supplement 2*). All booster immunizations induced strong class 5 escape peaks targeting RBD residues 352, 357, 396, and 468 (*Figure 4C and D*). Importantly, the bivalent Wu1/BA.5 S-EABR mRNA-LNP booster induced a less pronounced overall class 5 profile than the other boost immunogens, which

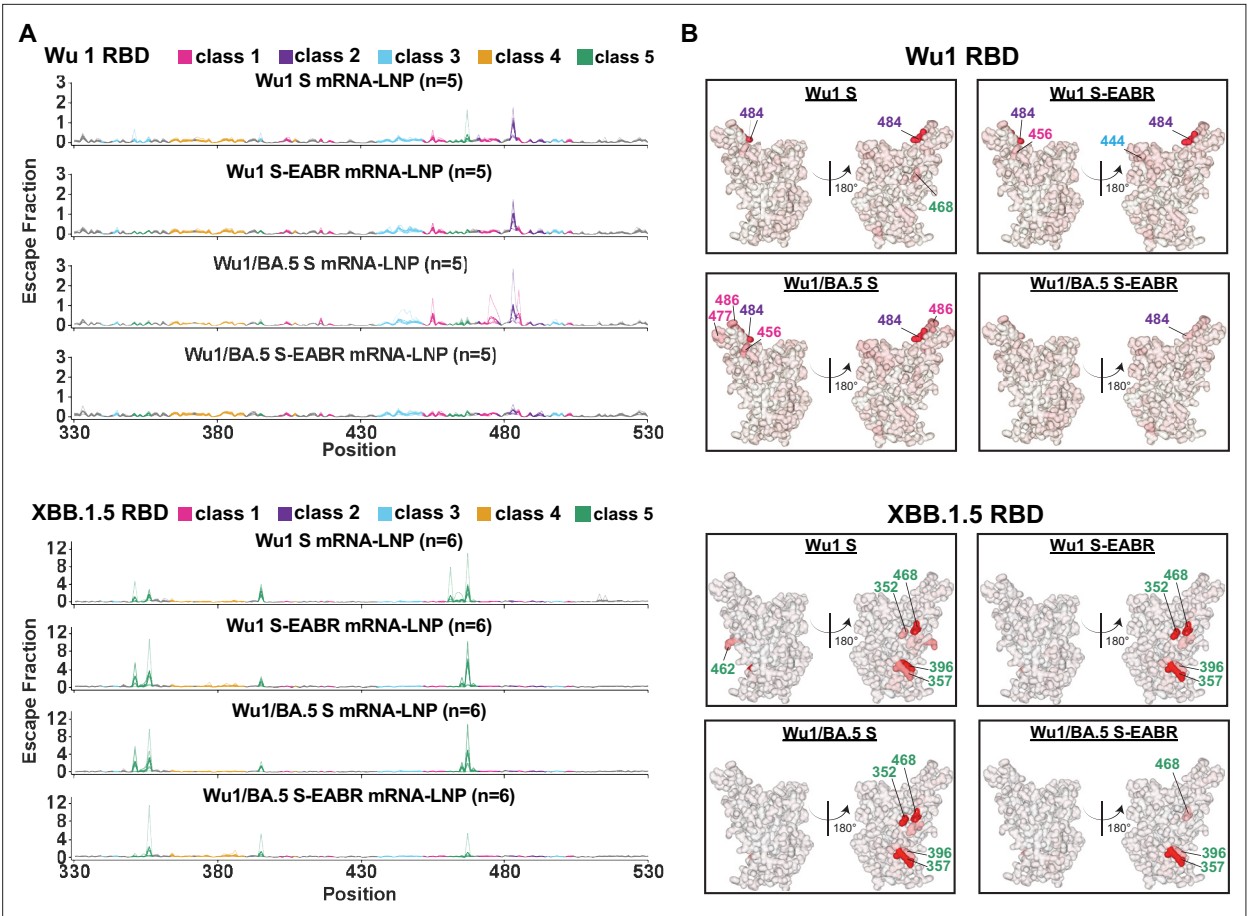

**Figure 4.** Bivalent S-EABR mRNA-LNP booster induces polyclass antibody (Ab) responses. Sera for deep mutational scanning (DMS) studies were derived from pre-vaccinated mice that received booster immunizations with 2 μg of monovalent Wu1 S mRNA-LNP, monovalent Wu1 S-EABR mRNA-LNP, bivalent Wu1/BA.5 S mRNA-LNP, or bivalent Wu1/BA.5 S-EABR mRNA-LNP (immunization schedule in *Figure 1A and B*). DMS was performed for 6 serum samples for each cohort, which were selected based on XBB.1 neutralization activity (1st, 2nd, 3rd, 5th, 6th, and 10th best neutralizer from each cohort). For the Wu1 receptor-binding domain (RBD) library, only five DMS profiles were obtained for each cohort. (**A**) Line plots for DMS results using Wu1 (top) or XBB.1.5 (bottom) RBD libraries for sera from pre-vaccinated mice that received the indicated boost immunogens. The x axes display the RBD residue numbers, and the y axes show the sum of the Ab escape of all mutations at a site (larger numbers indicating increased Ab escape). Each faint line shows one antiserum with heavy lines representing the average across the n=5 (top) or n=6 (bottom) sera in each cohort. Lines are colored differently for RBD epitopes within different RBD epitope classes (*Barnes et al., 2020*; *Fan et al., 2022*) (color definitions shown in legend above line plots; epitopes are shown in *Figure 1—figure supplement 2*; gray color for residues not assigned to an epitope). (**B**) The average site-total Ab escape for the Wu1 (top) or XBB.1.5 (bottom) RBD libraries for pre-vaccinated mice that received the indicated boost immunogens mapped to the Wu1 RBD surface (PDB: 6M0J). Gray coloring indicates no escape, and red coloring indicates escape. Lines with numbers colored according to epitope class indicate RBD positions with the most escape.

The online version of this article includes the following figure supplement(s) for figure 4:

**Figure supplement 1.** Bivalent S-EABR mRNA-LNP booster induces polyclass antibody (Ab) responses against Wu1 receptor-binding domain (RBD).

**Figure supplement 2.** Bivalent S-EABR mRNA-LNP booster induces less pronounced class 3 Ab responses against XBB.1.5 receptor-binding domain (RBD).

was primarily driven by a strong class 5 response observed in a single animal in this cohort (*Figure 4—figure supplement 2*; *Supplementary file 2*). The other mice that received the bivalent Wu1/BA.5 S-EABR mRNA-LNP booster exhibited polyclass or weak to moderate responses against class 4 and 5 epitopes. In contrast, strong class 5 profiles were detected in 4–5 mice in cohorts that received the monovalent Wu1 S or S-EABR mRNA-LNP boosters or the bivalent Wu1/BA.5 S mRNA-LNP booster (*Figure 4—figure supplement 2*; *Supplementary file 2*). Overall, these results suggest that the bivalent Wu1/BA.5 S-EABR mRNA-LNP booster induced a more diversified polyclass Ab response that

likely contributed to the higher neutralizing activity observed against Omicron subvariants, which could result in more effective recognition of future variants.

## Co-expression of Wu1 and BA.5 S results in heterotrimer formation

The results from this and previous studies *Scheaffer et al., 2023*; *Khoury et al., 2023*; *Chalkias et al., 2022*; *Collier et al., 2023* demonstrated that bivalent booster immunizations improve the potency and breadth of Ab responses against Omicron subvariants compared to monovalent boosters. Serum depletion assays also showed that the BA.5 neutralizing activity elicited by the bivalent Wu1/BA.5 S and S-EABR mRNA-LNP boosters was primarily driven by cross-reactive Abs that bind to the Wu1 and BA.5 RBDs (*Figure 3—figure supplement 3*). Preferential activation of cross-reactive B cells could be achieved through co-display of Wu1 and BA.5 S trimers on the surface of a cell and/or an eVLP (*Figure 1C*), which could result in inter-S crosslinking between cross-reactive B cell receptors (BCRs) and neighboring homotrimeric Wu1 and BA.5 S trimers (*Figure 5—figure supplement 1*). As an additional mechanism for bivalent booster-induced activation of cross-reactive B cells, we hypothesized that co-expression of ancestral Wu1 and Omicron S in the same cell could result in the formation of S heterotrimers consisting of Wu1 and Omicron S protomers and could promote intra-S crosslinking, i.e., simultaneous binding of cross-reactive BCRs to two neighboring RBDs within the same S trimer (*Figure 5—figure supplement 1*). We previously postulated that intra-S crosslinking is possible for several anti-RBD IgGs based on modeling of intact IgG coordinates into single-particle cryo-electron microscopy (cryo-EM) structures of fragment antigen-binding (Fab)-S complexes (*Barnes et al., 2020*; *Fan et al., 2022*), and intra-S crosslinking was observed in EM studies of IgG-S trimer complexes (*Callaway et al., 2023*). However, it is currently unknown whether co-expression of Wu1 and Omicron Ss induces heterotrimerization, and if so, whether heterotrimer formation affects the overall structure of the S ectodomain and influences bivalent booster-induced immune responses.

To investigate whether S heterotrimers could form with different S protomers, we designed 6P versions (containing 6 proline substitutions to enhance expression and stability) (*Hsieh et al., 2020*) of Wu1/BA.1 and Wu1/BA.1/Delta soluble heterotrimer Ss (HT2 and HT3, respectively) by co-transfection with specific C-terminally-tagged constructs. HT2 was generated by co-transfecting StrepII-tagged Wu1 S and His-tagged BA.1 S, while HT3 was generated using StrepII-tagged Wu1 S, His-tagged BA.1 S, and D7324 (*Sanders et al., 2013*)-tagged Delta S (*Figure 5A*). Each heterotrimer was subsequently purified using sequential affinity purification specific to each tag, confirming the inclusion of distinct protomers in the purified heterotrimers (*Figure 5A*).

HT2 and HT3 S proteins were then evaluated using single-particle cryo-EM to determine whether heterotrimeric Ss resembling their homotrimeric counterparts were formed. To ensure that C-terminal tags did not disrupt the formation of trimers, we included the following controls: (i) three homotrimers with single tags (StrepII-tagged Wu1 S, His-tagged BA.1 S, and D7324-tagged Delta S) and (ii) a homotrimer comprised of His- and StrepII-tagged protomers (Wu1-His/Wu1-StrepII). Single-particle cryo-EM analysis confirmed that all control S proteins formed trimers, indicating that C-terminal tags did not disrupt trimerization (*Figure 5—figure supplement 2*). Because most of the EM constructions for homotrimeric S controls exhibited preferred orientations, we determined overall conformational states (*Figure 5*) for the control structures but did not fit atomic models. Structural analysis also revealed trimerized HT2 and HT3 Ss (*Figure 5B and C*; *Figure 5—figure supplement 2*). Although atomic models could not be built for the HT2 and HT3 trimer reconstructions because the identities of individual protomers (Wu1 or BA.1 for HT2, and Wu1, BA.1, or Delta for HT3) could not be conclusively determined at relatively low resolution, the EM densities of S heterotrimers and control homotrimers showed no major conformational differences in domain organization (*Figure 5B and C*; *Figure 5—figure supplement 2*). As previously seen in SARS-CoV-2 S trimer structures (*Walls et al., 2020*; *Wrapp et al., 2020*), we observed two trimer populations in all datasets—one with all three RBDs in the 'down' conformation and one with a single RBD in an 'up' conformation (*Figure 5D*). These data demonstrate heterotrimeric S formation for soluble forms of SARS-CoV-2 S proteins. This result suggests that the bivalent S or S-EABR mRNA-LNP boost immunogens likely induce heterotrimerization, which could promote activation of cross-reactive B cells through intra-S crosslinking, a potential mechanism warranting further investigation.

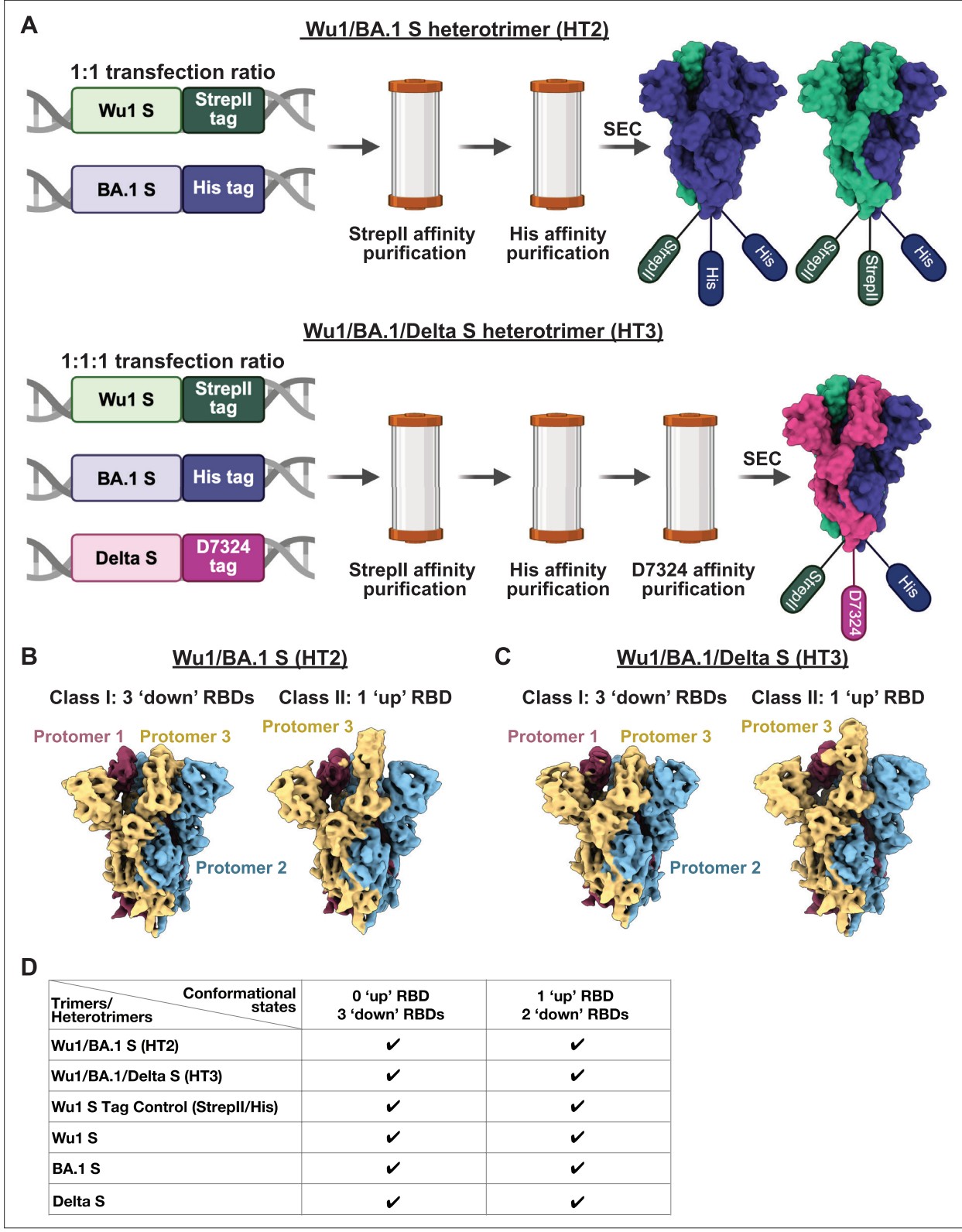

**Figure 5.** Heterotrimer S formation occurs for soluble constructs. (**A**) Schematic for the generation and purification of soluble SARS-CoV-2 Wu1/BA.1 (HT2) and Wu1/BA.1/Delta (HT3) S heterotrimers. HT2 heterotrimers were generated by co-transfection of StrepII-tagged Wu1 S and His-tagged BA.1 S constructs at a 1:1 DNA ratio, while HT3 heterotrimers were produced by co-transfecting StrepII-tagged Wu1 S, His-tagged BA.1 S, and D7324-tagged Delta S constructs at a 1:1:1 ratio. Each heterotrimer was sequentially purified using affinity chromatography specific for each tag (StrepII, His, or JR-52) to ensure incorporation of distinct protomers. This panel was created using BioRender.com. (**B**) Cryo-electron microscopy (cryo-EM) density maps for

*Figure 5 continued on next page*

*Figure 5 continued*

Wu1/BA.1 HT2 S class I with 3 'down' receptor-binding domains (RBDs) (3.8 Å resolution) and class II with 1 'up' RBD (3.8 Å resolution). (**C**) 4.0 Å and 3.9 Å resolution cryo-EM density maps for Wu1/BA.1/Delta HT3 S with 3 'down' RBDs and 1 'up' RBD. (**D**) Table summarizing S heterotrimer and control homotrimer conformational states.

The online version of this article includes the following figure supplement(s) for figure 5:

**Figure supplement 1.** Cartoon diagram illustrating inter- and intra-S crosslinking.

**Figure supplement 2.** Cryo-electron microscopy (cryo-EM) data collection and processing for heterotrimeric and control SARS-CoV-2 S structures.

# Discussion

The EABR mRNA vaccine platform delivers mRNA encoding engineered immunogens that induce budding of eVLPs from the plasma membrane to promote presentation of immunogens on cell surfaces and eVLPs (*Hoffmann et al., 2023*). This approach elicited superior neutralizing Ab responses in naïve mice compared to a conventional mRNA vaccine against original and variant SARS-CoV-2 (*Hoffmann et al., 2023*). However, the majority of the human population has already mounted immune responses against SARS-CoV-2 through prior vaccinations, natural infections, or a combination of both. Here, we demonstrate that the EABR mRNA vaccine approach improves humoral immune responses elicited by monovalent and bivalent mRNA-LNP booster immunizations against Omicron subvariants in pre-vaccinated mice.

The possibility of receiving a third dose of the original COVID-19 mRNA vaccine in fall 2021 (*Andrews et al., 2022b*) coincided with the emergence of the Omicron variants in November 2021 (*WHO, 2021*). These variants had acquired many substitutions, especially in the RBD, which allowed efficient evasion of Ab-mediated immunity (*Liu et al., 2022*; *Cele et al., 2022*) and rapid global spread. Similar to the effects of a third vaccination in humans (*Gruell et al., 2022*), a third immunization in pre-vaccinated mice with a monovalent Wu1 S mRNA-LNP booster elicited modest neutralizing activity against the early Omicron variants BA.1 and BA.5. However, the addition of an EABR sequence (*Hoffmann et al., 2023*) to the boosting immunogen resulted in higher Ab neutralizing titers against BA.1 and BA.5, suggesting that a monovalent Wu1 S-EABR booster could have promoted enhanced protection against early Omicron variants. Importantly, the monovalent Wu1 S-EABR mRNA-LNP booster elicited similar neutralizing responses against the BA.5, BQ.1.1, and XBB.1 subvariants compared to the conventional bivalent S mRNA-LNP booster that contained both Wu1 and BA.5 S immunogens.

While updating of mRNA vaccines can be done relatively quickly, the development, clinical testing, approval, and large-scale manufacturing of a bivalent booster that includes updated immunogens based on new circulating variants cannot be accomplished fast enough to keep up with viral evolution during a global pandemic such as COVID-19. Indeed, the Wu1/BA.5 bivalent booster became available in fall 2022, almost 1 year after the emergence of Omicron variants (*Song et al., 2024*). The results reported here suggest that the EABR mRNA vaccine approach presents a potential solution that could reduce the need for frequent development of updated COVID-19 boosters and potentially future pandemic viruses. Combined with our previous finding that an initial two-shot vaccination series with a Wu1 S-EABR mRNA-LNP immunogen elicited substantially higher neutralizing activity against Omicron variants than conventional Wu1 S mRNA-LNP immunizations in naïve mice (*Hoffmann et al., 2023*), our results demonstrate that the EABR mRNA vaccine approach has the potential to induce more effective and lasting humoral immunity against emerging SARS-CoV-2 variants and represents a promising platform for future pandemic preparedness.

Clinical studies showed that the bivalent COVID-19 mRNA booster outperformed the monovalent mRNA booster in terms of preventing symptomatic illness, hospitalizations, and deaths when administered as a fourth dose in humans (*Song et al., 2024*; *Tan et al., 2023*). Consistent with other studies (*Scheaffer et al., 2023*; *Khoury et al., 2023*; *Chalkias et al., 2022*; *Collier et al., 2023*; *Davis-Gardner et al., 2023*), the bivalent conventional and EABR-based boosters evaluated in this study elicited higher neutralizing responses against Omicron subvariants than their monovalent counterparts in pre-vaccinated mice, although improvements were relatively modest. However, the bivalent Wu1/BA.5 S-EABR mRNA-LNP booster consistently elicited the highest RBD binding and neutralizing titers against all tested Omicron subvariants and induced robust neutralizing responses in the highest number of animals against the BQ.1.1 and XBB.1 variants that are particularly difficult

to neutralize (*Wang et al., 2023c*; *Miller et al., 2023*). These results highlight that the EABR mRNA vaccine approach and the bivalent booster strategy work synergistically to induce more effective humoral immune responses.

In addition to eliciting the most potent neutralizing responses against Omicron subvariants, the bivalent Wu1/BA.5 S-EABR mRNA-LNP booster also induced distinct DMS signatures compared to the other boost immunogens, consistent with differences in epitope mapping of polyclonal anti-RBD Ab responses. While the monovalent Wu1 S and S-EABR mRNA-LNP and the bivalent Wu1/BA.5 S mRNA-LNP boosters elicited predominantly class 1 and class 2 responses against Wu1 RBD and strong class 5 responses against XBB.1.5 RBD, boost immunizations with bivalent Wu1/BA.5 S-EABR mRNA-LNP elicited polyclass serum Ab responses against Wu1 RBD and mostly polyclass or moderate polyclass class 4 and class 5 responses against XBB.1.5 RBD. These results imply that combining the bivalent booster strategy with the EARB mRNA vaccine approach elicited a more diversified Ab response, resulting in simultaneous and balanced targeting of multiple RBD epitopes. A more diverse anti-RBD Ab response might be desirable as viral immune evasion would require mutations in multiple sites, including the more conserved class 4 and 5 RBD epitopes. As a potential mechanism for the more frequent induction of polyclass-like Ab responses, we hypothesize that EABR eVLP formation promotes close proximity of Wu1 and BA.5 S trimers on densely coated eVLPs, consistent with cryo-electron tomography imaging of purified eVLPs showing that the density of S trimers was equivalent to or higher than that on SARS-CoV-2 virions (20–40 S trimers on 40–60 nm diameter eVLPs) (*Hoffmann et al., 2023*). The high density of co-displayed Wu1 and BA.5 S trimers on eVLPs should facilitate inter-S crosslinking by BCRs that recognize epitopes that are conserved between Wu1 and BA.5 S trimers, leading to preferential activation of cross-reactive B cells. By comparison, conventional bivalent mRNA booster-mediated co-expression of Wu1 and BA.5 S trimers could result in lower S trimer density on cell surfaces that are crowded with endogenous membrane proteins, which could prevent efficient inter-S crosslinking.

As an additional mechanism for bivalent booster-induced preferential activation of cross-reactive B cells, we provide evidence that co-expression of ancestral and Omicron S results in heterotrimer formation that could promote intra-S crosslinking, i.e., simultaneous binding of both Fabs of a cross-reactive BCR to neighboring RBDs within the same S trimer. We previously postulated that intra-S crosslinking is possible for several anti-RBD Abs that target various RBD epitopes based on modeling of intact IgG structures onto cryo-EM structures of Fab-S complexes (*Barnes et al., 2020*; *Fan et al., 2022*). The conventional mRNA- and the EABR mRNA-based bivalent boost immunogens should promote similar degrees of heterotrimerization and intra-S crosslinking. While our RBD serum depletion assays showed that both bivalent boosters primarily elicited cross-neutralizing Abs that bind to the Wu1 and BA.5 RBDs, only the EABR mRNA-based bivalent booster induced predominantly polyclass DMS signatures. Thus, it is possible that the diversification of the anti-RBD Ab response observed for the bivalent EABR mRNA booster was primarily driven by inter-S crosslinking possibly in conjunction with intra-S crosslinking. Future studies are needed to investigate the impact of inter- and intra-S crosslinking on the activation of cross-reactive B cells and to evaluate whether the EABR mRNA vaccine approach can improve other types of bivalent and multivalent vaccines.

Although our results highlight the potential of the EABR mRNA vaccine approach in the context of pre-existing immunity, we observed that monovalent and bivalent EABR mRNA booster-induced humoral immune responses were still largely dictated by immunological imprinting resulting from the initial vaccination series. Post-boost BA.5 neutralizing responses were roughly an order of magnitude lower than Wu1 neutralization titers, even for bivalent boosters that included BA.5 S immunogens. In addition, despite eliciting consistent polyclass anti-RBD Ab responses, bivalent EABR mRNA booster-induced neutralizing activity against the BQ.1.1 and XBB.1 variants dropped over 10-fold compared to BA.5 neutralization. Notably, Ab binding responses to BQ.1.1 and XBB.1 RBDs were similar compared to BA.5 RBD binding for all boost immunogens. This discrepancy between RBD binding responses and neutralization titers indicates that only a relatively small fraction of binding Abs has neutralizing activity and/or the Abs still bind to the BQ.1.1 and XBB.1 RBDs, but fail to effectively prevent interactions with ACE2 on the surface of target cells. The initial two-shot COVID-19 mRNA vaccination series and all booster immunizations elicited potent binding responses against the S2 region, which is relatively conserved among SARS-CoV-2 variants (*Li and Chang, 2023*). Consistent with previous reports (*Bowen et al., 2022*), our depletion studies showed that anti-S2 Abs had little impact on virus

neutralization, which was instead predominantly driven by anti-RBD Abs. However, anti-S2 Abs could still provide valuable immune protection against symptomatic Omicron infections through Fc-mediated effector functions (*Balinsky et al., 2023*). In addition, mRNA vaccines induce potent cytotoxic T cell responses that cross-recognize Omicron variants (*Keeton et al., 2022*) and reduce COVID-19 disease severity (*Rydyznski Moderbacher et al., 2020*).

Taken together, our results demonstrate that the EABR mRNA vaccine approach improves monovalent and bivalent booster-mediated humoral immune responses against Omicron subvariants in prevaccinated mice. DMS epitope mapping studies further showed that the bivalent EABR mRNA booster elicited a more diversified anti-RBD Ab response than all other tested boost immunogens. Our work also provides evidence that bivalent mRNA booster immunizations promote heterotrimer formation, which could activate cross-reactive B cells through intra-S crosslinking. Despite these advances, future work is needed to develop booster strategies that overcome immunological imprinting more effectively and confer lasting immunity against emerging variants.

## Limitations of the study

Due to the substantial time required to conduct immunization studies in pre-vaccinated mice, this proof-of-concept study to evaluate the potential of EABR-modified boosters in the context of pre-existing immunity was performed using outdated SARS-CoV-2 variants. Future studies are needed to assess EABR booster responses against currently circulating variants. While the monovalent Wu1 S-EABR mRNA-LNP booster elicited robust improvements in neutralization titers against BA.1 and BA.5 compared to the conventional Wu1 S mRNA-LNP booster, consistent with previously reported results in naïve mice (*Hoffmann et al., 2023*), only modest and statistically nonsignificant increases were observed for later Omicron variants, such as BQ.1.1 and XBB.1. Moreover, although the bivalent Wu1/BA.5 S-EABR mRNA-LNP booster consistently induced the highest neutralizing titers across all tested variants, these increases were statistically significant only relative to the conventional monovalent Wu1 S mRNA-LNP and not the bivalent Wu1/BA.5 S mRNA-LNP booster. Larger mouse cohorts could have increased the statistical power of the study, such that the observed trends might have reached statistical significance. Another factor contributing to the relatively modest improvements elicited by the monovalent and bivalent EABR boosters against BQ.1.1 and XBB.1 was the generally low levels of neutralizing responses observed against these variants. Although the low neutralization activity against BQ.1.1 and XBB.1 was likely the result of immune imprinting caused by the initial vaccination series, addition of more cohorts might have excluded the possibility that the ancestral Wu1 S immunogen is more immunogenic than the Omicron BA.5 S immunogen, e.g., three immunizations with monovalent BA.5 S or bivalent Wu1/BA.5 S mRNA-LNP. However, this is unlikely since prior studies reported that ancestral and Omicron S immunogens are similarly immunogenic (*Scheaffer et al., 2023*; *Muik et al., 2022*; *Ying et al., 2022*). Finally, it remains unclear why the polyclonal RBD epitope-targeting response elicited by the bivalent Wu1/BA.5 S-EABR mRNA-LNP booster translated into only modest gains in neutralizing activity against Omicron variants. It is possible that a longer post-boost follow-up period or an additional booster incorporating an XBB.1 S immunogen would have revealed further maturation and expansion of these responses, and thereby additional benefits of the bivalent EABR booster.

## Methods
### Design of EABR constructs

SARS-CoV-2 S-EABR constructs were designed as described (*Hoffmann et al., 2023*). The EABR domain (residues 160–217) of the human CEP55 protein was fused to the C-terminus of the SARS-CoV-2 S protein obtained from the Wu1 (GenBank: QTA38991.1) or BA.5 (GenBank: UPP14409.1) variants separated by a 4-residue (Gly)$_3$Ser (GS) linker (*Figure 1—figure supplement 1A*). These constructs contained the native furin cleavage site, 2P stabilizing mutations (*Pallesen et al., 2017*), and the C-terminal 21 residues were truncated to remove an endoplasmic reticulum retention signal (*McBride et al., 2007*). The D614G substitution (*Korber et al., 2020*), naturally present in BA.5, was inserted into Wu1 S and S-EABR constructs. To prevent endocytosis and enhance the cell surface expression of S-EABR constructs, an EPM (residues 243–290 of mouse FcgRII-B1) (*Hoffmann et al.,*

*2023*; *Miettinen et al., 1989*) was inserted upstream of the 4-residue Gly-Ser linker and the EABR domain as described (*Hoffmann et al., 2023*; *Figure 1—figure supplement 1A*).

## In vitro transfection

To verify that the designed constructs induce eVLP budding, HEK293T cells were transiently transfected with DNA plasmids encoding Wu1 or BA.5 S-EABR using Fugene HD (Promega). After 48 hr, transfected cell supernatants were harvested, and eVLPs were purified by ultracentrifugation on a 20% sucrose cushion as described (*Hoffmann et al., 2023*). The purified eVLP fractions were 3-fold serially diluted in 0.1 M $NaHCO_3$ pH 9.6, directly coated onto Costar high-binding 96-well plates (Corning), and eVLP-associated S protein levels were quantified by ELISA. After 1 hr blocking in Tris-buffered saline with 0.1% Tween 20 (TBS-T) and 3% bovine serum albumin (BSA) (TBS-T/3% BSA) at room temperature, the monoclonal anti-S2 Ab S2P6 (*Pinto et al., 2021*) was diluted to a concentration of 5 µg/mL in TBS-T/3% BSA and added to plates for 1 hr at room temperature. After washing with TBS-T, HRP-conjugated goat anti-human IgG (2015-05; SouthernBiotech) was diluted 1:8000 in TBS-T/3% BSA and added to plates for 30 min at room temperature. After washing with TBS-T, plates were developed using 1-Step Ultra TMB-ELISA Substrate Solution (Thermo Fisher Scientific), and absorbance was measured at 450 nm.

## mRNA synthesis

Codon-optimized mRNAs encoding Wu1 S, Wu1 S-EABR, BA.5 S, and BA.5 S-EABR constructs were synthesized by RNAcore (https://www.houstonmethodist.org/research-cores/rnacore/) using proprietary manufacturing protocols as described (*Hoffmann et al., 2023*). In brief, mRNAs were synthesized by T7 RNA polymerase-mediated in vitro transcription using DNA templates encoding the immunogen open reading frame flanked by 5′ and 3′ untranslated regions and terminated by an encoded polyA tail. CleanCap 5′ cap structures (TriLink) were co-transcriptionally incorporated into the 5′ end. Uridine was completely substituted with N1-methyl-pseudouridine to lower immunogenicity (*Karikó et al., 2008*). mRNAs were purified by oligo-dT affinity purification and high-performance liquid chromatography to remove double-stranded RNA contaminants (*Karikó et al., 2011*). Purified mRNAs were stored at –80°C.

## LNP encapsulation of mRNAs

Purified mRNAs were formulated in LNP as described (*Hoffmann et al., 2023*; *Pardi et al., 2015*). In brief, 1,2-distearoyl-*sn*-glycero-3-phosphocholine, cholesterol, a PEG lipid, and an ionizable cationic lipid dissolved in ethanol were rapidly mixed with an aqueous acidic solution containing the mRNAs using an in-line mixer. The ionizable lipid and LNP composition are described in the international patent application WO2017075531(2017). The post in-line solution was dialyzed with PBS to remove ethanol and displace the acidic solution. Subsequently, mRNA-LNPs were measured for size (60–65 nm) and polydispersity (PDI<0.075) by dynamic light scattering (Malvern Nano ZS Zetasizer). The encapsulation efficiencies were >97% as measured by the Quant-iT Ribogreen Assay (Invitrogen).

## Immunizations

All animal procedures in this study were performed in compliance with the U.S. Department of Agriculture's (USDA) Animal Welfare Act (9 CFR Parts 1, 2, and 3), the Guide for the Care and Use of Laboratory Animals (Institute of Laboratory Animal Resources, National Academy Press, Washington, DC, 2011), and the National Institutes of Health, Office of Laboratory Animal Welfare. Whenever possible, procedures in this study were designed to avoid or minimize discomfort, distress, and pain to animals. Animals were observed for signs of morbidity and/or mortality at least once daily, in accordance with the Animal Welfare Act and the Association for Assessment and Accreditation of Laboratory Animal Care International (AAALACi). All mouse procedures were approved by the Labcorp Institutional Animal Care and Use Committee (IACUC).

7- to 8-week-old female BALB/c mice (Charles River Laboratories) were purchased and housed at Labcorp Drug Development, Denver, PA, for immunization studies. All animals were healthy upon receipt and were weighed and monitored during a 7-day acclimation period before the start of the study. Mice were randomly assigned to experimental cohorts of 10 animals. Up to 10 mice were

co-housed together, and cages were kept in a climate-controlled room at 63–77°C at 50 ± 20% relative humidity. Mice were provided Rodent Diet #5001 (Purina Lab Diet) ad libitum.

2 μg of Wu1 S mRNA-LNP were administered to all cohorts of mice by IM injections on days 0 and 28. On day 230, pre-vaccinated mice received different boosters by IM injection: (i) 2 μg Wu1 S mRNA-LNP; (ii) 2 μg Wu1 S-EABR mRNA-LNP; (iii) 1 μg Wu1 S mRNA-LNP+1 μg BA.5 S mRNA-LNP; (iv) 1 μg Wu1 S-EABR mRNA-LNP+1 μg BA.5 S-EABR mRNA-LNP. For bivalent boosters (iii and iv), the Wu1 and BA.5 S or S-EABR mRNA-LNP were mixed prior to the IM injections. Serum samples for ELISAs and neutralization assays were obtained on indicated days by retro-orbital bleeding. One small drop of either 0.5% proparacaine hydrochloride ophthalmic solution or 0.5% tetracaine hydrochloride ophthalmic solution was applied in the eye of the mouse prior to performing retro-orbital blood collection. Eyes were alternated for each blood collection. Euthanasia was performed in accordance with the American Veterinary Medical Association (AVMA) guidelines for euthanasia. Mice were individually placed in a $CO_2$ chamber and rendered unconscious. Mice were verified as unconscious by lack of response when the toe was pinched. Unconscious mice were subjected to bilateral jugular vein severance for terminal exsanguination. Mice were placed back in the $CO_2$ chamber until death occurred. A double thoracotomy was performed as a second confirmation of death.

## Protein expression and purification

Soluble SARS-CoV-2 S trimers (Wu1/D614G, BA.5, BQ.1.1, XBB.1) (*Hsieh et al., 2020*), RBDs (Wu1, BA.5, BQ.1.1, XBB.1), and a prefusion-stabilized S2 protein (*Hsieh et al., 2024*) were expressed as described (*Wang et al., 2022*; *Cohen et al., 2022*). Briefly, Avi/His-tagged S trimers and His-tagged RBD and S2 proteins were purified from transiently transfected Expi293F cells (A14527; Thermo Fisher; authenticated by supplier; negative for mycoplasma) by nickel affinity chromatography (HisTrap, Cytiva) and size-exclusion chromatography (SEC) (Superose 6 Increase 10/300, Cytiva) (*Barnes et al., 2020*; *Wang et al., 2022*; *Cohen et al., 2022*). Peak fractions corresponding to S, RBD, or S2 proteins were pooled, concentrated, and stored at 4°C. Biotinylated S trimers for ELISAs were generated by co-transfection of Avi/His-tagged S constructs with a plasmid encoding an endoplasmic reticulum-directed BirA enzyme (kind gift from Michael Anaya, Caltech).

For heterotrimeric constructs, Wu1/BA.1 (HT2) and Wu1/BA.1/Delta (HT3) heterotrimers were expressed via the addition of C-terminal tags and co-transfection of Expi293F cells (Gibco) with specific plasmid DNA ratios. HT2 heterotrimers were produced by co-transfecting a 1:1 plasmid DNA ratio of StrepII-tagged Wu1 S and His-tagged BA.1 S constructs. HT3 heterotrimers were produced by co-transfecting a 1:1:1 plasmid DNA ratio of StrepII-tagged Wu1 S, His-tagged BA.1 S, and D7324 (*Sanders et al., 2013*)-tagged Delta S constructs. HT2 was sequentially purified using StrepTrap (Cytiva), HisTrap (Cytiva), and SEC, while HT3 also underwent immunoaffinity chromatography using the D7324-specific monoclonal Ab JR-52 (*Sanders et al., 2013*) (kind gift from James Robinson, Tulane University) before SEC.

The Wu1-His/Wu1-StrepII S homotrimer was generated by co-transfecting Expi293F cells with a 1:1 plasmid DNA ratio of His-tagged Wu1 S and StrepII-tagged Wu1 S, followed by purification as described for the HT2 heterotrimer. Single-tagged S homotrimers included StrepII-tagged Wu1 S, His-tagged BA.1 S, and D7324-tagged Delta S, each purified using their respective affinity chromatography steps followed by SEC.

## Single-particle cryo-EM sample preparation

Grids of SARS-CoV-2 S homotrimers and heterotrimers were prepared at a protein concentration of 2 mg/mL. Prior to freezing, fluorinated octylmaltoside solution (Anatrace) was added to the S proteins at a final concentration of 0.02%(wt/vol). Three μL of S proteins were immediately applied to Quantifoil 300 mesh 1.2/1.3 grids (Electron Microscopy Sciences) that were freshly glow-discharged with the PELCO easiGLOW (Ted Pella) for 60 s at 20 mA. Grids were blotted for 3 s using Whatman No.1 filter paper with 0 blot force, 100% humidity, and room temperature, and vitrified in 100% liquid ethane using a Vitrobot Mark IV system (Thermo Fisher Scientific).

## Single-particle cryo-EM data collection and processing

Single-particle cryo-EM datasets for SARS-CoV-2 S homotrimers and heterotrimers were collected using SerialEM automatic data collection software (*Mastronarde, 2005*) on a 300 keV Titan Krios

(Thermo Fisher Scientific) cryo-electron microscope equipped with a K3 direct electron detector camera (Gatan). Movies were acquired in 40 frames with a defocus range of –1 to –3 µm and a total dosage of 60 electrons per $Å^2$ using the 3 by 3 beam image shift pattern with 3 exposures per hole in the super-resolution mode at a pixel size of 0.416 Å. Data processing workflows are outlined in *Figure 5—figure supplement 2*. For all datasets, motion correction was done with patch motion correction with a binning factor of 2, and contrast transfer function (CTF) parameters were estimated with Patch CTF in cryoSPARC (*Punjani et al., 2017*). Particles with a diameter of 100–200 Å were picked with a blob picker, inspected, extracted, and finally classified with 2D classification in cryo-SPARC (*Punjani et al., 2017*). Ice particles, junk particles, and particles with preferred orientations were discarded, and the remaining particles were used for ab initio reconstruction of three to four volumes. These reconstructions were further refined with heterogeneous refinement in cryoSPARC (*Punjani et al., 2017*). Depending on the state for the RBDs in the reconstructions, particles were combined for homogeneous and nonuniform refinements to obtain the final reconstructions in cryo-SPARC (*Punjani et al., 2017*).

## ELISAs

ELISAs were performed as described (*Hoffmann et al., 2023*; *Cohen et al., 2024*). To analyze serum binding to S trimers, pre-blocked streptavidin-coated Nunc MaxiSorp 384-well plates (Thermo Fisher Scientific) were coated with 5 µg/mL biotinylated S trimers in TBS-T/3% BSA for 1 hr at room temperature. To analyze serum binding to RBDs or S2, Nunc MaxiSorp 384-well plates (Sigma) were coated with 2.5 µg/mL RBD or S2 in 0.1 M $NaHCO_3$ pH 9.8 and incubated overnight at 4°C. Plates were blocked with TBS-T/3% BSA for 1 hr at room temperature followed by removal of the blocking buffer by aspiration. Serum samples from immunized mice were diluted 1:100, 4-fold serially diluted in TBS-T/3% BSA, and added to plates. After a 3 hr incubation at room temperature, plates were washed with TBS-T using an automated plate washer. HRP-conjugated goat anti-mouse IgG (715-035-150; Jackson ImmunoResearch) was diluted 1:100,000 in TBS-T/3% BSA and added to plates for 1 hr at room temperature. After washing with TBS-T, plates were developed using SuperSignal ELISA Femto Maximal Signal Substrate (Thermo Fisher Scientific), and absorbance was measured at 425 nm. Area under the curve calculations for binding curves were performed using GraphPad Prism 9.3.1 assuming a one-site binding model with a Hill coefficient as described (*Cohen et al., 2021*).

## Neutralization assays

Lentivirus-based SARS-CoV-2 pseudoviruses were generated as described (*Crawford et al., 2020*; *Robbiani et al., 2020*) using S proteins from the Wu1/D614G, BA.1, BA.5, BQ.1.1, and XBB.1 variants in which the C-terminal 21 residues of the S cytoplasmic tails were removed (*Crawford et al., 2020*). Serum samples from immunized mice were heat-inactivated for 30 min at 56°C. Three-fold serial dilutions of heat-inactivated samples were incubated with pseudoviruses for 1 hr at 37°C, followed by the addition of the serum-virus mixtures to pre-seeded HEK293T-ACE2 target cells. After 48 hr incubation at 37°C, BriteLite Plus substrate (Perkin Elmer) was added, and luminescence was measured. Half-maximal inhibitory dilutions ($ID_{50}$s) were calculated using 4-parameter nonlinear regression analysis in AntibodyDatabase (*West et al., 2013*), and $ID_{50}$ values were rounded to three significant figures.

## Ab depletions

HiTrap NHS-activated resin (Cytiva) was coupled to purified SARS-CoV-2 Wu1 RBD or a mock protein (HIV-1 gp120) at a 10 mg/mL concentration and stored in PBS supplemented with 0.02% sodium azide at 4°C. For Ab depletions, 50 µL dry worth of RBD-coupled resin were aliquoted into microcentrifuge tubes and spun down at 13,000 × $g$ for 2 min. The supernatant was discarded. The resin was washed with 100 µL of sterile PBS to remove excess PBS-azide solution. The supernatant was again discarded. 10 µL of mouse serum was then diluted in 90 µL PBS and added to the resin. The tubes were mixed and incubated at room temperature for 10 min. The resin-serum mixture was spun down at 13,000 × $g$ for 7 min to allow the resin to settle, and 90 µL of the supernatant was transferred into a fresh microcentrifuge tube. To ensure no resin was present in the final sample, the collection tube was spun down again, and supernatant was transferred to a fresh microcentrifuge tube. Sterile PBS was added for a final volume of 200 µL. Prior to neutralization assays, the Ab-depleted serum samples were heat-inactivated at 56°C for 30 min.

## Deep mutational scanning

DMS studies to map RBD epitopes recognized by serum Abs were performed in biological duplicates using independent mutant RBD yeast libraries (Wu1 [*Starr et al., 2020*; *Starr et al., 2022*] and XBB.1.5 RBD [*Taylor and Starr, 2023*] libraries were generously provided by Tyler Starr, University of Utah) as described (*Cohen et al., 2024*; *Cohen et al., 2022*; *Greaney et al., 2021a*; *Hills et al., 2024*). Briefly, yeast libraries were stained with serum Abs, labeled with a secondary Ab (115-605-008, Jackson ImmunoResearch), and then processed using a Sony SH800 cell sorter. Cells were gated to capture RBD mutants that had reduced Ab binding for a relatively high degree of RBD expression. Ab-escaped cells were expanded for DNA extraction, and Illumina sequencing was done as described (*Cohen et al., 2024*; *Greaney et al., 2021a*; *Hills et al., 2024*). Raw sequencing data are available on the NCBI SRA under BioProject PRJNA1067836. Escape fractions were computed using previously described processing steps (*Cohen et al., 2024*; *Greaney et al., 2021a*; *Hills et al., 2024*) and implemented using a Swift DMS program available from authors upon request.

Escape map visualizations shown as static line plots, logo plots, and structural depictions were all created using Swift DMS as previously described (*Fan et al., 2025*; *Cohen et al., 2024*; *Hills et al., 2024*). In some visualizations, RBD sites were categorized based on epitope region, which were defined as class 1 (pink) (RBD residues 403, 405, 406, 417, 420, 421, 453, 455–460, 473–478, 486, 487, 489, 503, 504), class 2 (purple) (residues 472, 479, 483–485, 490–495), class 3 (blue) (residues 341, 345, 346, 437–450, 496, 498–501), class 4 (orange) (residues 365–390, 408) (*Barnes et al., 2020*), or class 5 (green) (352–357, 396, 462–468) (*Fan et al., 2025*). Structural visualizations were performed using an RBD surface (PDB 6M0J) colored by the site-wise escape metric at each site as described (*Cohen et al., 2024*).

## Statistical analysis

Statistically significant titer differences between immunized cohorts of BALB/c mice (10 mice per cohort) for ELISAs were determined using analysis of variance (ANOVA) test followed by Tukey's multiple comparison post hoc test calculated using GraphPad Prism 9.3.1. Titer differences between immunized cohorts of BALB/c mice for Wu1, BA.5, BQ.1.1, and XBB.1 neutralization assays were determined using the non-parametric Kruskal-Wallis test followed by Dunn's multiple comparison post hoc test calculated using GraphPad Prism 9.3.1. For BA.1 neutralization titers, statistically significant differences were determined using the non-parametric two-tailed Mann-Whitney test calculated using GraphPad Prism 9.3.1.

## Acknowledgements

We thank J Vielmetter, A Lam, and the Caltech Protein Expression Center for assistance with protein production, S Chen and the Caltech Cryo-EM facility for cryo-EM data collection, M Anaya (Caltech) for a BirA expression plasmid, JE Robinson (Tulane University) for the JR-52 antibody, J Bloom (Fred Hutchinson Cancer Research Center) and T Starr (University of Utah) for RBD libraries, and J Bloom (Fred Hutchinson) and P Bieniasz (Rockefeller University) for neutralization assay reagents. We thank J Keeffe for guidance on ELISA method development, A West for Swift DMS support, Y Tam (Acuitas Therapeutics) for careful reading of the manuscript, B Wold, G Tolomiczenko, and the Caltech Merkin Institute for Translational Research for helpful discussions. Cryo-EM was performed in the Beckman Institute Resource Center for Transmission Electron Microscopy at Caltech. We thank Labcorp Drug Development–Antibody Reagents and Vaccines (Denver, PA) (formerly Covance, Inc) for mouse immunization studies, and RNAcore (Houston Methodist Research Institute) for synthesis of mRNAs and helpful discussion. These studies were funded by Wellcome Leap (PJB), the National Institutes of Health P01-AI165075 (PJB) and DP5OD033362 (MAGH), Gates Foundation INV-034638 (PJB) and INV-056219 (MAGH), the Merkin Institute for Translational Research (Caltech) (PJB), and the George Mason University Fast Grants (PJB). This manuscript is the result of funding in whole or in part by the National Institutes of Health (NIH). It is subject to the NIH Public Access Policy. Through acceptance of this federal funding, NIH has been given a right to make this manuscript publicly available in PubMed Central upon the Official Date of Publication, as defined by NIH.

## Additional information

### Competing interests

Woohyun J Moon: employee of Acuitas Therapeutics, a company developing lipid nanoparticle delivery technology. Paulo JC Lin: employee of Acuitas Therapeutics, a company developing lipid nanoparticle delivery technology; holds equity in Acuitas Therapeutics. Pamela J Bjorkman: inventor on a US patent application filed by the California Institute of Technology that covers the EABR technology described in this work; is a scientific advisor for Vaccine Company, Inc. Magnus AG Hoffmann: inventor on a US patent application filed by the California Institute of Technology that covers the EABR technology described in this work; is a scientific consultant for Vaccine Company, Inc. The other authors declare that no competing interests exist.

### Funding

| Funder | Grant reference number | Author |
|---|---|---|
| National Institutes of Health | P01-AI165075 | Pamela J Bjorkman |
| National Institutes of Health | DP5OD033362 | Magnus AG Hoffmann |
| Gates Foundation | INV-034638 | Pamela J Bjorkman |
| Gates Foundation | INV-056219 | Magnus AG Hoffmann |
| Merkin Institute for Translational Research, California Institute of Technology | | Pamela J Bjorkman |
| George Mason University | | Pamela J Bjorkman |
| Wellcome Leap | | Pamela J Bjorkman |

The funders had no role in study design, data collection and interpretation, or the decision to submit the work for publication. For the purpose of Open Access, the authors have applied a CC BY public copyright license to any Author Accepted Manuscript version arising from this submission.

### Author contributions

Chengcheng Fan, Kim-Marie A Dam, Conceptualization, Data curation, Formal analysis, Validation, Investigation, Visualization, Methodology, Writing – original draft, Writing – review and editing; Alexander A Cohen, Conceptualization, Data curation, Formal analysis, Supervision, Validation, Investigation, Visualization, Methodology, Writing – original draft, Writing – review and editing; Annie V Rorick, Ange-Célia I Priso Fils, Priyanthi NP Gnanapragasam, Kathryn E Huey-Tubman, Data curation, Formal analysis, Investigation, Methodology; Zhi Yang, Data curation, Formal analysis, Validation, Investigation, Methodology; Luisa N Segovia, Data curation, Investigation, Methodology; Woohyun J Moon, Resources, Methodology; Paulo JC Lin, Resources, Methodology, Writing – review and editing; Pamela J Bjorkman, Conceptualization, Resources, Formal analysis, Supervision, Funding acquisition, Investigation, Visualization, Methodology, Writing – original draft, Project administration, Writing – review and editing; Magnus AG Hoffmann, Conceptualization, Resources, Data curation, Formal analysis, Supervision, Funding acquisition, Validation, Investigation, Methodology, Writing – original draft, Project administration, Writing – review and editing

### Author ORCIDs

Chengcheng Fan ⓘ https://orcid.org/0000-0003-4213-5758
Alexander A Cohen ⓘ https://orcid.org/0000-0002-2818-656X
Kim-Marie A Dam ⓘ https://orcid.org/0000-0002-1416-4757
Zhi Yang ⓘ https://orcid.org/0000-0001-8680-3784
Pamela J Bjorkman ⓘ https://orcid.org/0000-0002-2277-3990
Magnus AG Hoffmann ⓘ https://orcid.org/0000-0003-4923-9568

## Ethics

All animal procedures in this study were performed in compliance with the U.S. Department of Agriculture's (USDA) Animal Welfare Act (9 CFR Parts 1, 2, and 3); the Guide for the Care and Use of Laboratory Animals (Institute of Laboratory Animal Resources, National Academy Press, Washington, D.C., 2011); and the National Institutes of Health, Office of Laboratory Animal Welfare. Whenever possible, procedures in this study were designed to avoid or minimize discomfort, distress, and pain to animals. Animals were observed for signs of morbidity and/or mortality at least once daily, according to animal welfare act and the Association for Assessment and Accreditation of Laboratory Animal Care International (AAALACi). All mouse procedures were approved by the Labcorp Institutional Animal Care and Use Committee (IACUC).

Reviewer #1 (Public review): https://doi.org/10.7554/eLife.108959.3.sa1
Reviewer #3 (Public review): https://doi.org/10.7554/eLife.108959.3.sa2
Author response https://doi.org/10.7554/eLife.108959.3.sa3

---

# Additional files

## Supplementary files

Supplementary file 1. Geometric mean neutralization titers. Day 56, 154, and 244 geometric mean neutralization titers for cohorts 1–4 against the Wu1, BA.5, BQ.1.1, and XBB.1 variants.

Supplementary file 2. Bivalent S-EABR mRNA-LNP booster induces consistent polyclass anti-receptor-binding domain (RBD) antibody (Ab) responses. The deep mutational scanning (DMS) profiles for antisera from individual mice against the Wu1 and XBB.1.5 RBD libraries shown in *Figure 4—figure supplements 1–2* were classified as polyclass (total escape peaks<1 at all sites; green), moderate escape (total escape peaks between 1 and 2 at one or more sites; yellow), or strong escape (total escape peaks >2 at one or more sites; red) profiles. Total escape peaks represent the sum of escape fractions for all mutations at a specific site (*Greaney et al., 2021b*). For moderate and strong escape profiles, the RBD epitope classes with the highest total escape peaks are specified.

MDAR checklist

## Data availability

Code used for data processing and visualization of DMS results is available at data.caltech.edu (DOI: https://doi.org/10.22002/j4k4w-p8331). DMS raw sequencing data is available under an NCBI SRA (BioProject: PRJNA1067836; BioSample: SAMN56741139; SRA: SRX32770151). Cryo-EM density maps are deposited in the EMDB with accession codes: EMD-71513, EMD-71514, EMD-71515 and EMD-71516. Materials are available upon request to corresponding authors with a signed material transfer agreement, and other information required to analyze the data in this paper is available from the lead contacts upon request. This paper does not report original code.

The following datasets were generated:

| Author(s) | Year | Dataset title | Dataset URL | Database and Identifier |
|---|---|---|---|---|
| California Institute of Technology | 2026 | SARS-CoV-2 RBD mutational antig Illumina barcode sequencing for XBB15_22_gp8_81_250_abneg | https://www.ncbi.nlm.nih.gov/sra/SRX32770151 | NCBI Sequence Read Archive, SRX32770151 |
| Fan C, Dam KA, Bjorkman PJ | 2025 | SARS-CoV-2 Wuhan-Hu-1/BA.1 spike heterotrimer (HT2) with 3-'down' RBDs | https://emdataresource.org/EMD-71513 | EMDataBank, EMD-71513 |
| Fan C, Dam KA, Bjorkman PJ | 2025 | SARS-CoV-2 Wuhan-Hu-1/BA.1 spike heterotrimer (HT2) with 1-'up' RBD | https://emdataresource.org/EMD-71514 | EMDataBank, EMD-71514 |

*Continued on next page*

*Continued*

| Author(s) | Year | Dataset title | Dataset URL | Database and Identifier |
|---|---|---|---|---|
| Fan C, Dam KA, Bjorkman PJ | 2025 | SARS-CoV-2 Wuhan-Hu-1/BA.1/Delta spike heterotrimer (HT3) with 3-'down' RBDs | https://emdatasource.org/EMD-71515 | EMDataBank, EMD-71515 |
| Fan C, Dam KA, Bjorkman PJ | 2025 | SARS-CoV-2 Wuhan-Hu-1/BA.1/Delta spike heterotrimer (HT3) with 1-'up' RBD | https://emdatasource.org/EMD-71516 | EMDataBank, EMD-71516 |

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
