## [Editor Report · eLife Assessment]

This **useful** and interesting study provides evidence that EABR mRNA is at least as effective as standard S mRNA vaccines for SARS-CoV-2. The authors provide **convincing** justification for the conclusion that the inconsistent statistical significance for Omicron is likely due to immune imprinting or original antigenic sin. In this regard, the significance of the findings is stronger as it points to possible challenges for updated vaccine strategies in overcoming immune imprinting.

---

## [Referee Report · Reviewer #1 (Public review)]

Summary:

This study investigated the immunogenicity of a novel bivalent EABR mRNA vaccine for SARS-CoV-2 that expresses enveloped virus-like particles in pre-immune mice as a model for boosting the population that is already pre-immune to SARS-CoV-2. The study builds on promising data showing a monovalent EABR mRNA vaccine induced substantially higher antibody responses than a standard S mRNA vaccine in naïve mice. In pre-immune mice, the EABR booster increased the breadth and magnitude of the antibody response, but for Omicron, the effects were modest and often not statistically significant. The authors provide compelling evidence to support this may be due to immune imprinting.

This study also builds on prior work with additional experiments to elucidate the mechanisms that contributed to the EABR increased immunogenicity in naive mice including evidence that the vaccine is inducing responses to more RBD epitopes and a potential role for heterodimer formation as a mechanism whereby bivalent vaccines induce cross-reactive B cell responses.

Strengths:

Evaluating a novel SARS-CoV-2 vaccine that was substantially superior in naive mice in pre-immune mice as a model for its potential in the pre-immune population.

Providing insight into a possible role of immune imprinting in shaping immune responses to updated booster immunizations.

Minor weaknesses:

(1) Overall, immune responses against Omicron variants were substantially lower than against the ancestral Wu-1 strain that the mice were primed with. The authors speculate this is evidence of immune imprinting. While parallel controls (mice immunized 3 times with just the bivalent EABR vaccine) were not tested, the authors point to prior published work showing Omicron S antigen is a strong immunogen. This indicates the lower immune responses to Omicron are likely due to immune imprinting (or original antigenic sin) and not due to S immunogen being inherently less immunogenic than the S protein from the ancestral Wu-1 strain.

(2) The authors reported statistically significant increase in antibody responses with the bivalent EABR vaccine booster when compared to the monovalent S mRNA vaccine but consistently failed to show significantly higher responses when compared to the bi-valent S mRNA vaccine suggesting that in pre-immune mice, the EABR vaccine has no apparent advantage over the bivalent S mRNA vaccine which is the current standard. There were, however, some trends indicating the group sizes were insufficiently powered to see a difference. The discussion acknowledges these limitations of their studies and potential limited benefits of the EABR strategy in pre-immune mice vs standard bivalent mRNA vaccine.

(3) The EABR S mRNA vaccine was superior to the conventional mRNA S vaccine in naïve mice but not in pre-immune mice. The authors expanded the discussion to propose a possible role for immune imprinting in this result which is supported by the data.

---

## [Referee Report · Reviewer #3 (Public review)]

Summary:

The authors evaluated a novel bivalent (Wu1/BA.5 based) mRNA platform that uses the EABR strategy to produce enveloped virus-like particles for vaccination. These were tested as boosters in the context of pre-existing immunity in mice that received two prior immunizations with conventional Wu1 mRNA vaccines. The animal experimental timeline aimed at mimicking the vaccinations/booster schedule implemented during the COVID-19 pandemia. The authors tested and compared different booster strategies: (1) conventional Wu1 S protein encoding mRNA vaccine, (2) EABR Wu1 S protein encoding mRNA vaccine that produces enveloped virus-like particles, (3) conventional Wu1/BA.5 S protein encoding mRNA vaccine, and (4) EABR Wu1/BA.5 S protein encoding mRNA vaccine that produces enveloped virus-like particles. The EABR approach (monovalent or bivalent) enhanced the antibody response against Wu1 and Omicron subvariants. Interestingly, the bivalent EABR Wu1/BA.5 mRNA (strategy 4) generated polyclonal sera targeting multiple receptor-binding domain epitopes: these sera were more diverse than those generated with the other tested booster strategies (1 to 3).

Strengths:

The monovalent Wu1 S-EABR mRNA booster led to increase in antibody binding to tested Omicron variants (BA.5, BQ.1.1, XBB.1), while the bivalent Wu1/BA.5 S-EABR mRNA booster led to the highest Ab response against Omicron variants (BA.5, BQ.1.1, XBB.1) in pre-vaccinated mice.

Neutralization assays showed that the monovalent Wu1 S-EABR mRNA booster had the highest Wu1 neutralization activity and to a lesser extent the early BA.1 early Omicron variant. The monovalent Wu1 S-EABR mRNA booster and bivalent Wu1/BA.5 S-EABR mRNA booster had similar BA.5 neutralizing activity. Neutralizing activity of the different boosters was less pronounced with later Omicron variants BQ.1.1 and XBB.1. However, of the different boosters tested, the bivalent Wu1/BA.5 S-EABR mRNA booster induced the highest neutralizing titers. These results support that the EABR mRNA vaccine strategy helps improve neutralizing activity against different tested Omicron subvariants: a few (1 or 2) mRNA constructs expressing major antigens in enveloped virus-like particles likely provide a novel strategy to elicit an immune response that has the potential to neutralize subsequent variants.

The EABR enveloped virus-like particle strategy induces a more diverse antibody response, including epitopes not recognized by the other booster strategies: these new epitopes could play a role in neutralizing activity against new future variants.

Moreover, the bivalent Wu1/BA.5 S-EABR mRNA booster could potentially produce heterotrimeric S proteins to help activation of cross-reactive B cells and increase polyclass antibody responses.

Weaknesses:

When it comes to later Omicron variants (BQ.1.1 and XBB.1), there is a discrepancy between epitope binding response and neutralization titers: only a few binding antibodies have neutralizing activity with these later variants, showing a limitation of the EABR strategy.

The authors showed that the EABR mRNA strategy represents a novel antigen exposing strategy where antigens are produced at the cell surface and also at the surface of enveloped virus-like particles. This allows the production of novel antigens in addition to those that would be typically generated against cell surface exposed antigens. These novel antigens targeting new epitopes could potentially have neutralizing activity.

Using a bivalent EABR mRNA booster led to higher antibody titers and higher neutralizing activity. The challenge is to select the best antigen target/variant to support neutralizing activity against later virus variants.

---

## [Author Response]

The following is the authors’ response to the original reviews.

**eLife Assessment**
This report provides useful evidence that EABR mRNA is at least as effective as standard S mRNA vaccines for the SARS-CoV-2 booster vaccine. Although the methodology and the experimental approaches are solid, the inconsistent statistical significance throughout the study presents limitations in interpreting the results. Also, the absence of results showing possible mechanisms underlying the lack of benefit with EABR in the pre-immune makes the findings mostly observational.

Thank you for your assessment of our study. Respectfully, we do not agree that our study shows a lack of benefit of using the EABR approach. For the monovalent boosters, the S-EABR mRNA booster improved neutralizing antibody titers by 3.4-fold against BA.1 (p = 0.03; Fig. S5) and 4.8-fold against BA.5 (failed to reach statistical significance; Fig. 3B) compared to the regular S mRNA booster, which is consistent with the findings from our prior study in naïve mice. In addition, the bivalent S-EABR booster consistently elicited the highest neutralizing titers against all tested variants, including significantly higher titers against BA.5 and BQ.1.1 than the monovalent S booster. The bivalent S-EABR booster also induced detectable neutralization activity in a larger number of mice than all other boosters.

Consistent with this analysis, please note that reviewers 1 and 2 commented that “the EABR booster increased the breadth and magnitude of the antibody response, but the effects were modest and often not statistically significant” (reviewer 1) and “the authors found that across both monovalent and bivalent designs, the EABR antigens had improved antibody titers than conventional antigens, although they observed dampened titers against Omicron variants, likely due to immune imprinting” (reviewer 2).

We agree with the reviewers’ assessment that the EABR booster-mediated improvements were mostly modest, in particular against the BQ.1.1 and XBB.1 strains. We also acknowledge that the improvements in titers did not reach statistical significance in many cases, which we believe could have been addressed by adding more animals to our cohorts. Unfortunately, that would have been prohibitively expensive and time-consuming given that we already included 10 mice per group, which is standard practice in the vaccine field.

Finally, we also wish to point out that we did include experiments that addressed potential mechanistic differences between booster groups. For example, we conducted deep mutational scanning studies to determine polyclonal antibody epitope mapping profiles, showing that bivalent S-EABR boosters induced more balanced targeting of multiple RBD epitopes, which likely contributed to the observed improvements in neutralization. Our work also included cryo-EM studies demonstrating that bivalent S mRNA boosters promote heterotrimer formation, which could potentially drive preferential stimulation of cross-reactive B cells via intra-spike crosslinking. This represents a potential mechanism explaining how bivalent boosters outperformed monovalent boosters in our and many prior studies, which warrants further investigation. Finally, we also performed serum depletion assays, showing that the BA.5 neutralizing activity elicited by the bivalent Wu1/BA.5 S and S-EABR mRNA boosters was primarily driven by cross-neutralizing Abs induced by the primary vaccination series.

**Public Reviews:**

**Reviewer #1 (Public review):**
Summary:This study investigated the immunogenicity of a novel bivalent EABR mRNA vaccine for SARS-CoV-2 that expresses enveloped virus-like particles in pre-immune mice as a model for boosting the population that is already pre-immune to SARS-CoV-2. The study builds on promising data showing a monovalent EABR mRNA vaccine induced substantially higher antibody responses than a standard S mRNA vaccine in naïve mice. In pre-immune mice, the EABR booster increased the breadth and magnitude of the antibody response, but the effects were modest and often not statistically significant.

We thank the reviewer for their accurate summary of our study. Please see our comments to the reviewer’s individual points below, as well as our responses to the editor’s assessment above.

Strengths:Evaluating a novel SARS-CoV-2 vaccine that was substantially superior in naive mice in pre-immune mice as a model for its potential in the pre-immune population.Weaknesses:(1) Overall, immune responses against Omicron variants were substantially lower than against the ancestral Wu-1 strain that the mice were primed with. The authors speculate this is evidence of immune imprinting, but don't have the appropriate controls (mice immunized 3 times with just the bivalent EABR vaccine) to discern this. Without this control, it's not clear if the lower immune responses to Omicron are due to immune imprinting (or original antigenic sin) or because the Omicron S immunogen is just inherently more poorly immunogenic than the S protein from the ancestral Wu-1 strain.

The reviewer raises an important point, and we agree that including additional groups receiving three immunizations with the bivalent spike and/or spike-EABR mRNA vaccines would have improved the experimental design. However, we believe that several prior studies have already demonstrated that Omicron S immunogens are not inherently poorly immunogenic compared to the ancestral S; e.g., Scheaffer et al., Nat Med (2022); Ying et al., Cell (2022); Muik et al., Sci Immunol (2022). Based on these prior reports, we conclude that the lower neutralizing titers against Omicron variants in our study are most likely driven by immune imprinting as a result of the initial vaccination series with the ancestral S immunogen.

(2) The authors reported a statistically significant increase in antibody responses with the bivalent EABR vaccine booster when compared to the monovalent S mRNA vaccine, but consistently failed to show significantly higher responses when compared to the bivalent S mRNA vaccine, suggesting that in pre-immune mice, the EABR vaccine has no apparent advantage over the bivalent S mRNA vaccine which is the current standard. There were, however, some trends indicating the group sizes were insufficiently powered to see a difference. This is mostly glossed over throughout the manuscript. The discussion section needs to better acknowledge these limitations of their studies and the limited benefits of the EABR strategy in pre-immune mice vs the standard bivalent mRNA vaccine.

We acknowledge that the improvements in titers did not reach statistical significance in many cases, which we believe could have been addressed by adding more animals to our cohorts. Unfortunately, that would have been prohibitively expensive and timeconsuming given that we already included 10 mice per group, which is standard practice in the vaccine field. We added a “Limitations of the study” section at the end of the discussion to address all of these points in detail (lines 570-598 in the revised version).

(3) The discussion would benefit from additional explanation about why they think the EABR S mRNA vaccine was substantially superior in naïve mice vs the standard S mRNA vaccine in their previously published work, but here, there is not much difference in pre-immune mice.

As we pointed out in our response to the editor’s assessment above, the monovalent SEABR mRNA booster improved neutralizing antibody titers by 3.4-fold against BA.1 (p = 0.03; Fig. S5) and 4.8-fold against BA.5 (failed to reach statistical significance; Fig. 3B) compared to the conventional monovalent S mRNA booster, which is largely consistent with the findings from our prior study in naïve mice. Although the bivalent S-EABR mRNA booster consistently elicited higher neutralizing titers than the conventional bivalent S mRNA booster, we agree with the reviewer that these improvements were modest and not statistically significant. Overall, neutralizing activity against later Omicron variants, such as BQ.1.1 and XBB.1 was low. We attributed this finding to immune imprinting (see response to point (1) above) and acknowledged that the EABR approach was not able to effectively overcome this effect (see discussion section of the paper, lines 537-558; and “Limitations of the study” section, lines 570-598 in the revised version).

**Reviewer #2 (Public review):**
Summary:In this manuscript, Fan, Cohen, and Dam et al. conducted a follow-up study to their prior work on the ESCRT- and ALIX-binding region (EABR) mRNA vaccine platform that they developed. They tested in mice whether vaccines made in this format will have improved binding/neutralization antibody capacity over conventional antigens when used as a booster. The authors tested this in both monovalent (Wu1 only) or bivalent (Wu1 + BA.5) designs. The authors found that across both monovalent and bivalent designs, the EABR antigens had improved antibody titers than conventional antigens, although they observed dampened titers against Omicron variants, likely due to immune imprinting. Deep mutational scanning experiments suggested that the improvement of the EABR format may be due to a more diversified antibody response. Finally, the authors demonstrate that co-expression of multiple spike proteins within a single cell can result in the formation of heterotrimers, which may have potential further usage as an antigen.

We thank the reviewer for their support and for the accurate summary and evaluation of our study.

Strengths:(1) The experiments are conducted well and are appropriate to address the questions at hand. Given the significant time that is needed for testing of pre-existing immunity, due to the requirement of pre-vaccinated animals, it is a strength that the authors have conducted a thorough experiment with appropriate groups.(2) The improvement in titers associated with EABR antigens bodes well for its potential use as a vaccine platform.Weaknesses:As noted above, this type of study requires quite a bit of initial time, so the authors cannot be blamed for this, but unfortunately, the vaccine designs that were tested are quite outdated. BA.5 has long been replaced by other variants, and importantly, bivalent vaccines are no longer used. Testing of contemporaneous strains as well as monovalent variant vaccines would be desirable to support the study.

We thank the reviewer for bringing up this important point. We agree that the variants used for this study are now outdated, and it would have been informative to evaluate conventional and EABR boosters against contemporaneous strains. However, as the reviewer correctly pointed out, this type of study requires a substantial amount of time to conduct and will therefore will likely always be outdated by the time the data are analyzed and prepared for publication. To accurately assess immune responses against recent or current strains in mice, multiple boosters would have been needed to mimic the pre-existing immune context in the human population in 2025. Assuming intervals of 6-7 months between boosters (as used in this study to mimic booster intervals in the human population as closely as possible), this type of study would have been challenging to conduct, especially given the limited lifespan of mice. Thus, we performed this proof-of-concept study using outdated variants to assess the potential of EABR-modified boosters. We greatly appreciate the reviewer’s understanding and acknowledge this limitation of our study, which is highlighted in the added “Limitations of the study” section in the revised version of the manuscript (lines 570-598).

**Recommendations for the authors:**

**Reviewer #1 (Recommendations for the authors):**
(1) The acronym RBD in the title should be spelled out.

We thank the reviewer for raising this point. We made this change in the revised version of the paper.

(2) Lines 167-168 describe no differences between the cohorts at day 244. It should also be stated that for all timepoints, there are no significant differences.

We modified the revised manuscript according to the reviewer’s suggestion (line 170).

**Reviewer #2 (Recommendations for the authors):**
(1) Given the focus on developing broad vaccines for future coronavirus outbreaks, it would be particularly informative to test whether the EABR antigens elicit broadened/heightened responses against other (beta)coronaviruses. If enough serum is left, it would seem straightforward to conduct neutralization assays against non-SARSCoV-2 coronaviruses.

We thank the reviewer for this valid suggestion. Unfortunately, the extensive analysis of the serum samples, including spike and RBD ELISAs and neutralization assays against multiple variants, deep mutational scanning, and depletion assays, used up the serum samples for most mice. We agree that it would be interesting to investigate whether bivalent EABR boosters elicit pan-sarbecovirus responses in future studies.

(2) In the bar plots for antibody titer changes, shown as log10 fold change, it is quite hard to interpret the difference between bars (e.g., what is the fold change difference between each bar in the same time point?). A table of mean {plus minus} SD values would be helpful.

That’s a great suggestion. We added a table (Table S1) presenting all the geometric mean neutralization titers for all timepoints and variants in the revised version of the manuscript.

(3) The development of heterotrimers as potential antigens is very interesting, but it seems out of place in the current manuscript. This should likely be in a separate, standalone manuscript.

We thank the reviewer for commenting on the heterotrimer part of our manuscript. The presented work was not intended to advance the development of heterotrimers as potential antigens. Instead, our findings demonstrate that bivalent spike mRNA vaccines readily generate heterotrimers, which could promote intra-spike crosslinking and potentially impact antibody epitope targeting profiles as suggested by the deep mutational scanning data for the bivalent S-EABR mRNA booster (Fig. 4; Fig. S7-8). We think this is an important consideration that warrants further investigation with regards to the development of future bivalent or multivalent vaccines.

(4) As a minor note, the sequences of the variants used or accession numbers should be provided in the Methods, since different groups have used different mutations for variants.

We added the accession numbers for the vaccine strains used in this study (lines 604605).